# Intra-islet α-cell Gs signaling promotes glucagon release

Liu Liu [1] ✉, Kimberley El [2], Diptadip Dattaroy[1], Luiz F. Barella [1], Yinghong Cui[1], Sarah M. Gray[2], Carla Guedikian [2], Min Chen[3], Lee S. Weinstein[3], Emily Knuth[4], Erli Jin[4], Matthew J. Merrins[4], Jeffrey Roman[5], Klaus H. Kaestner[5], Nicolai Doliba[5], Jonathan E. Campbell [2] & Jürgen Wess [1] ✉

Glucagon, a hormone released from pancreatic α-cells, is critical for maintaining euglycemia and plays a key role in the pathophysiology of diabetes. To stimulate the development of new classes of therapeutic agents targeting glucagon release, key α-cell signaling pathways that regulate glucagon secretion need to be identified. Here, we focused on the potential importance of α-cell $G_s$ signaling on modulating α-cell function. Studies with α-cell-specific mouse models showed that activation of α-cell $G_s$ signaling causes a marked increase in glucagon secretion. We also found that intra-islet adenosine plays an unexpected autocrine/paracrine role in promoting glucagon release via activation of α−cell $G_s$-coupled $A_{2A}$ adenosine receptors. Studies with α-cell-specific $Gα_s$ knockout mice showed that α-cell $G_s$ also plays an essential role in stimulating the activity of the *Gcg* gene, thus ensuring proper islet glucagon content. Our data suggest that α-cell enriched $G_s$-coupled receptors represent potential targets for modulating α-cell function for therapeutic purposes.

Pancreatic α-cells, which are contained within the islets of Langerhans, store and release glucagon, a 29 amino acid peptide hormone that plays a key role in regulating glucose homeostasis and various other metabolic functions[1–5]. Numerous studies have shown that glucagon release is dysregulated in diabetes and that this deficit contributes to the pathogenesis of diabetes[6–8]. In type 1 diabetes (T1D), impaired glucagon release in response to low blood glucose levels (e.g., after an insulin injection) can lead to severe hypoglycemia and even death[2,7]. In type 2 diabetes (T2D), plasma glucagon levels are inappropriately high, thus contributing to the hyperglycemia phenotype that is the major hallmark of T2D[6–8].

Two major actions of glucagon are to stimulate hepatic glucose production to maintain blood glucose levels under fasting conditions and to counteract the effects of insulin on the liver and other organs or cell types[2,9,10]. In addition, more recent studies suggest that glucagon released from pancreatic α-cells can act in a paracrine fashion on adjacent β-cells to promote the release of insulin when glucose levels are high (e.g., after a meal)[11–15]. This glucagon effect is mediated to a major extent by activation of glucagon-like peptide-1 receptors (GLP-1Rs) that are expressed at relatively high levels by pancreatic β-cells[11–15].

Taken together, these findings suggest that glucagon can have multiple effects on blood glucose homeostasis and that the nature of these effects depends on specific nutritional and other physiological conditions. These observations raise the possibility that drugs capable of modulating glucagon release from pancreatic α-cells or glucagon receptor (GCGR) agonists or antagonists may prove useful for the treatment of various metabolic disorders. In fact, several GCGR/GLP-1R dual agonists are currently undergoing clinical trials for the treatment of diabetes, obesity, and nonalcoholic steatohepatitis[1,16–18]. Preclinical studies suggest that GCGR/GLP-1R/glucose-dependent insulinotropic

[1]Molecular Signaling Section, LBC, National Institute of Diabetes and Digestive and Kidney Diseases, Bethesda, MD 20892, USA. [2]Duke Molecular Physiology Institute, Duke University, Durham, NC 27701, USA. [3]Metabolic Diseases Branch, National Institute of Diabetes and Digestive and Kidney Diseases, Bethesda, MD 20892, USA. [4]Division of Endocrinology, Diabetes and Metabolism, Department of Medicine, University of Wisconsin-Madison, Madison, WI 53705, USA. [5]Institute for Diabetes, Obesity and Metabolism, Perelman School of Medicine, University of Pennsylvania, Philadelphia, PA 19104, USA. ✉e-mail: liu.liu@nih.gov; jurgenw@niddk.nih.gov

polypeptide (GIP) receptor triple agonists may also prove useful for the therapy of diabetes and obesity[1,16,19,20].

While glucose is a key determinant of glucagon secretion[21–23], considerable gaps exist in our knowledge of the extracellular signals and α-cell signaling pathways that regulate glucagon release. However, a detailed understanding of these mechanisms may lead to the identification of new strategies aimed at modulating glucagon secretion for therapeutic purposes.

Like other cell types, pancreatic α-cells express dozens of GPCRs that are linked to different functional classes of heterotrimeric G proteins[24]. At present, the in vivo metabolic relevance of α-cell GPCRs that couple to the stimulatory G protein $G_s$ remains largely unknown, primarily due to the lack of suitable animal models. For this reason, the present study was designed to explore the role of α-cell $G_s$ signaling in regulating α-cell function and whole body glucose homeostasis by using several newly generated mouse models.

First, we generated a new mouse line that expressed a $G_s$-coupled designer receptor ($G_s$ DREADD or short GsD; DREADD=designer receptors exclusively activated by a designer drug)[25] selectively in pancreatic α-cells (α-GsD mice). This designer receptor, like other DREADD variants, is not recognized by endogenous ligands but can be selectively activated by small synthetic compounds such as clozapine-N-oxide (CNO) or deschloroclozapine (DCZ)[26–28]. When used in the proper dose or concentration range, CNO and DCZ are pharmacologically inert[27,29]. As a result, treatment of α-GsD mice with a DREADD agonist makes it possible to explore the outcome of selectively activating $G_s$ signaling in pancreatic α-cells in vivo and in vitro. Since essentially all GPCRs are expressed in multiple tissues and cell types[30], this has not been possible by using traditional pharmacological approaches.

During the course of these studies, we focused on an endogenous $G_s$-coupled receptor, the $A_{2A}$ adenosine receptor (A2AR), that is expressed selectively in both mouse and human α-cells, as compared to other cell types of the endocrine pancreas[31,32]. Studies with α-cell-selective A2AR knockout (KO) mice showed that α-cell A2ARs play a key role in promoting glucagon release when glucose levels are low.

Finally, we generated and analyzed a new mouse strain that lacked the gene coding for the α-subunit of $G_s$ ($G\alpha_s$) (gene name: *Gnas*) selectively in pancreatic α-cells of adult mice. Interestingly, the lack of α-cell $G_s$ signaling led to a significant reduction in islet glucagon content due to reduced transcription of the *Gcg* gene (encoding proglucagon), resulting in hypoglucagonemia and impaired glucagon secretion to a variety of stimuli.

In sum, systematic studies with several new mouse models demonstrated that α-cell $G_s$ signaling plays a central role in the control of α-cell function, including the regulation of *Gcg* expression and glucose homeostasis. These findings could pave the way towards the development of pharmacological agents that are able to modulate glucagon synthesis and/or release for various metabolic disorders.

## Results

### Selective expression of a $G_s$-coupled designer receptor in mouse α-cells

To explore the metabolic outcome of selectively activating $G_s$ signaling in pancreatic α-cells, we used DREADD technology to generate a mutant mouse strain that selectively expressed the Gs-DREADD (GsD) in α-cells of adult mice. Specifically, we intercrossed mice harboring the *GsD* allele preceded by a *loxP-STOP-loxP* (*LSL*) sequence (*CAG-LSL-GsD* mice)[33] with *Gcg-CreER^T2* mice[34] to generate heterozygous *CAG-LSL-GsD* mice containing one copy of the *Gcg-CreER^T2* transgene. These mutant mice were then treated with tamoxifen (TMX), thus promoting Cre activity and GsD expression in pancreatic α-cells and endocrine L cells of the gastrointestinal tract. Because intestinal L cells turn over rapidly, Cre-modified L-cells are known to be replaced by wild-type (WT) L-cells 4 weeks after TMX treatment[34]. Four weeks after the last

TMX injection, the *CAG-LSL-GsD Gcg-CreER^T2* mice expressed GsD selectively in α-cells (hereafter referred to as α-GsD mice). TMX-treated *CAG-LSL-GsD* mice that did not harbor the *Gcg-CreER^T2* transgene and thus did not express the GsD receptor served as control animals throughout the study.

The expression of the GsD designer receptor was detected using an antibody specific to the HA tag that had been fused to the N-terminus of GsD[33]. Immunoblotting analysis revealed that α-GsD mice expressed GsD in pancreatic islets but not in any other tissues including intestinal tissue known to contain proglucagon-producing L-cells (Supplementary Fig. 1a). Immunofluorescence staining of slices prepared from pancreatic islets from α-GsD mice confirmed the expression of GsD in glucagon-expressing α-cells and the lack of GsD expression in insulin-containing β-cells or other islet cells (Supplementary Fig. 1b). As expected, GsD was not detectable in pancreatic slices prepared from control mice (Supplementary Fig. 1b). Moreover, immunofluorescence staining of brain slices prepared from α-GsD mice failed to detect GsD expression in proglucagon-producing neurons of the nucleus tractus solitarius (NTS; Supplementary Fig. 1c). These observations indicate that GsD is selectively expressed in pancreatic α-cells of α-GsD mice.

### Stimulation of α-cell $G_s$ signaling leads to hyperglucagonemia and hyperinsulinemia in vivo

The expression of GsD in pancreatic α-cells did not affect body weight, islet size, or α- and β-cell mass (Supplementary Fig. 1d–g). α-GsD mice (males) and their control littermates consuming regular chow were then injected with either saline (i.p.) or the selective DREADD agonist, DCZ (10 μg/kg, i.p.)[27], followed by the monitoring of changes in plasma glucagon, plasma insulin, and blood glucose levels. Similar to previous observations[13,35], i.p. injection of control and α-GsD mice with saline or of control mice with DCZ resulted in reduced plasma glucagon levels 15 and 30 min after injection ($p < 0.0001$; one-way repeated measures ANOVA, followed by post-hoc Bonferroni adjustment) (Fig. 1a, d). Although the precise mechanism underlying this phenomenon remains unclear, it is possible that the stress of the i.p. injection is confounding basal hormonal levels. Although there was a trend towards higher basal plasma glucagon level in α-GsD mice (time 0; Fig. 1d), this effect failed to reach statistical significance ($p = 0.085$; Student's *t* test at time 0). Importantly, DCZ treatment of α-GsD mice led to a statistically significant increase in plasma glucagon levels (Fig. 1d), thus overcoming the inhibitory effect on glucagon release caused by the injection stress.

Saline treatment of α-GsD mice and control littermates resulted in statistically significant increases in blood glucose levels ($p < 0.0001$ at 15 and 30 min after injection (time 0); one-way repeated measures with time ANOVA for each group) (Fig. 1c). While this effect persisted in DCZ-treated control mice, blood glucose levels remained unchanged after DCZ injection of α-GsD mice (Fig. 1f), most likely due the increase in plasma insulin levels observed with DCZ-treated α-GsD mice (Fig. 1e) that "neutralized" the hyperglemic effect of the injection stress.

Acute DCZ (10 μg/kg, i.p.) treatment of α-GsD and control mice had no significant effect on the plasma levels of somatostatin and the two major incretin hormones, GIP and GLP-1 (note that GLP-1 is a cleavage product of proglucagon and is primarily secreted from intestinal L cells) (Supplementary Fig. 1h–j). These data support the concept that chemogenetic activation of α-cell $G_s$ signaling promotes the secretion of glucagon which can then act on adjacent β-cells to stimulate the release of insulin[11–15].

### Stimulation of α-cell $G_s$ signaling promotes glucagon release from perfused mouse islets

To confirm that the DCZ-induced changes in hormone secretion observed with α-GsD mice in vivo were indeed due to altered $G_s$

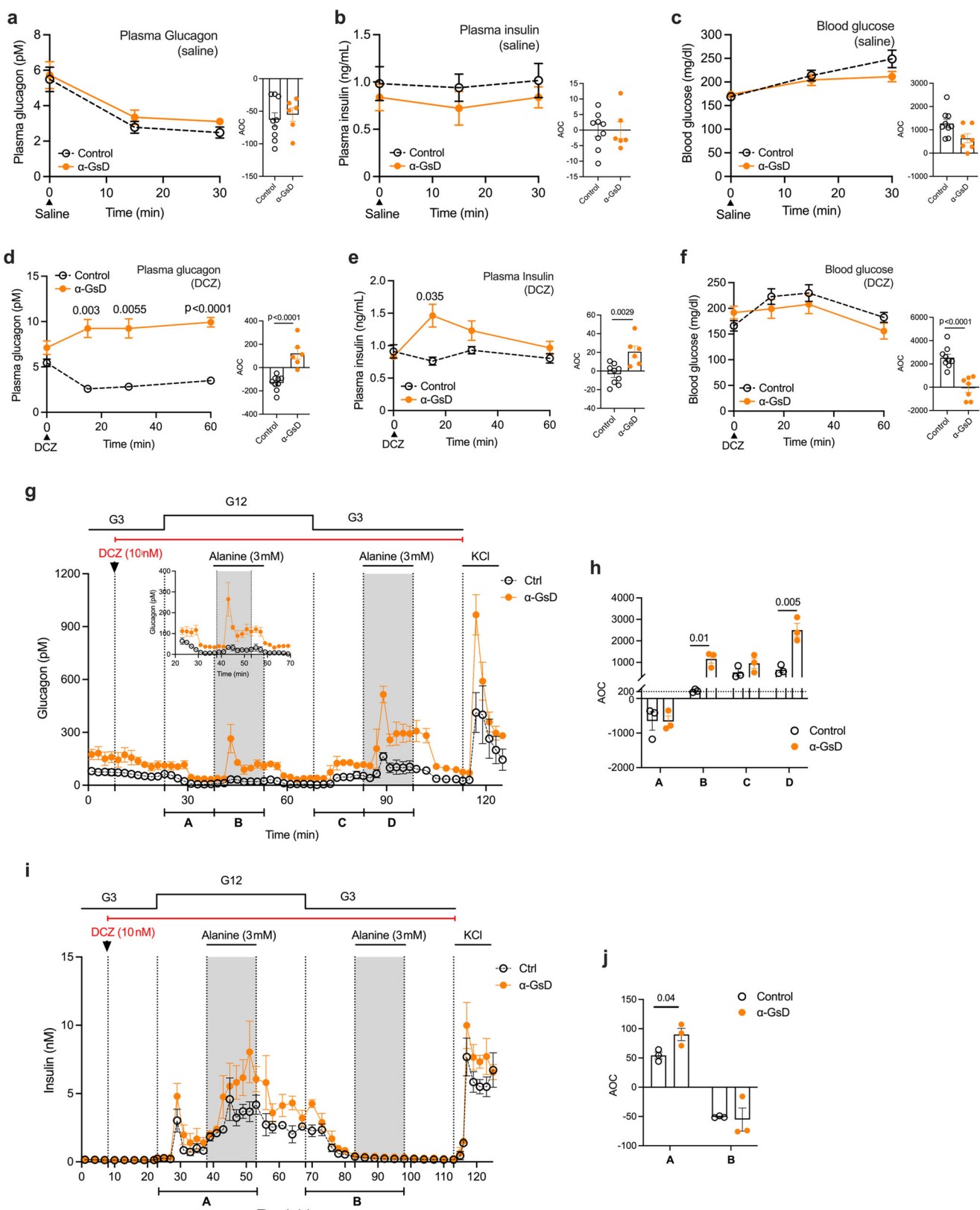

signaling in pancreatic α-cells, we conducted perifusion experiments using islets prepared from α-GsD mice (α-GsD islets) and control littermates (control islets). Basal glucagon release at both G3 and G12 was significantly higher in α-GsD islets, as compared to control islets (Fig. 1g, h and Supplementary Fig. 1k). This observation is consistent with previous findings that the GsD designer receptor shows a

certain degree of constitutive activity under distinct experimental conditions[25,33].

Basal insulin secretion was similarly low in both α-GsD and control islets at G3 (Supplementary Fig. 1l). Treatment of α-GsD and control islets with a physiological amino acid mixture (AAM) or alanine which stimulate the secretory activity of mouse α-cells[11,36] enhanced the

**Fig. 1 | Acute activation of the α-cell GsD designer receptor stimulates glucagon secretion.** Freely fed α-GsD mice and control littermates were injected with saline (**a**–**c**) or DCZ (10 μg/kg, i.p.) (**d**–**f**). Plasma glucagon (**a**, **d**), plasma insulin (**b**, **e**), and blood glucose (**c**, **f**) levels were measured at the indicated time points.
**g**–**j** Pancreatic islets prepared from control and α-GsD mice were perifused with the indicated glucose concentrations in the presence of DCZ (10 nM) and (alanine (3 mM)). Glucagon ((**g**); insert: glucagon from 23 to 70 min) and insulin secretion (**i**) were measured in the presence of low and high glucose levels (3 mM [G3] and 12 mM [G12], respectively). AOC values were calculated for glucagon (**h**) and insulin

(**j**) secretion calculated for different stimulation periods. All experiments were carried out with male littermates (12–16 weeks old). Data are given as means ± SEM (in vivo studies: control, $n = 9$; α-GsD, $n = 7$; in vitro studies: 3 independent perifusion experiments with 75–100 islets per perifusion chamber). Data were analyzed via two-way repeated measures ANOVA for time with Bonferroni post hoc test for comparison of individual time points (**d**–**f**) or two-tailed Student's $t$ test (**h**, **j**). Numbers of above data points or horizontal lines in the bar graphs represent p values. AOC, area of the curve. Source data are provided as a Source Data file.

secretion of both glucagon and insulin at G12 (Fig. 1g–j, Supplementary Fig. 1k, l), consistent with the concept that the paracrine effects of glucagon requires elevated glucose concentrations to stimulate insulin secretion[11–15]. DCZ (10 nM) treatment of α-GsD islets in the presence of 3 or 12 mM glucose resulted in significant increases in glucagon secretion only in the presence of alanine (3 mM) (Fig. 1g, h). To correct for differences in basal hormone secretion, we calculated area of the curve (AOC) values by subtracting the areas under or over the baseline[37]. This type of analysis is the method of choice when baseline levels between two or more experimental groups differ[37].

Under physiological conditions, pancreatic islets are exposed to high levels of circulating amino acids which increase the responsiveness of α-cells to various glucagon secretagogues[36,38], providing a likely explanation for the inability of DCZ to stimulate glucagon release from α-GsD islets in the absence of alanine (Fig. 1g, h). At G12, the DCZ/alanine-induced increases in glucagon release in α-GsD islets were accompanied by marked enhancements of glucose-stimulated insulin secretion (GSIS) (Fig. 1i, j). Thus, except for the enhanced basal activity of GsD in vitro, the islet perifusion data are in good agreement with the in vivo results described in the previous paragraph (Fig. 1d, e).

### Activation of α-cell Gs signaling improves glucose tolerance in both lean and obese mice

To explore the impact of acute activation of α-cell Gs signaling on glucose homeostasis, α-GsD mice and control littermates maintained on regular rodent chow (lean mice) were subjected to an i.p. glucose tolerance test (ipGTT). Following co-injection of DCZ (10 μg/kg, i.p.) and glucose (2 g/kg, i.p.), α-GsD mice showed a significant improvement in glucose tolerance, as compared to control littermates (Fig. 2a). This beneficial metabolic effect was associated with pronounced increases in both plasma glucagon and insulin levels in α-GsD mice (Fig. 2b, c), suggesting that the increase in insulin secretion following activation of α-cell Gs signaling causes improved glucose tolerance. Co-injection of control and α-GsD mice with insulin (0.75 IU/kg, i.p.) (insulin tolerance test, ITT) and DCZ (10 μg/kg; i.p.) resulted in comparable decreases in blood glucose levels in both groups of mice, indicating that stimulation of α-cell Gs signaling does not affect peripheral insulin sensitivity (Fig. 2d). In the absence of DCZ, control and α-GsD mice did not show any significant differences in blood glucose excursions in the ipGTT and ITT assays (Supplementary Fig. 2a, b). Taken together, these findings indicate that activation of α-cell Gs signaling leads to improved glucose tolerance, most likely due to increased insulin release triggered by enhanced glucagon secretion.

We next examined whether stimulation of α-cell Gs signaling also improved glucose homeostasis in obese, glucose-intolerant mice. To address this question, α-GsD mice and control littermates were maintained on a high-fat diet (HFD) for at least 8 weeks. Consumption of the HFD led to a similar degree of weight gain in both groups of mice (Supplementary Fig. 2c). The two cohorts of mice were then co-injected with glucose (1 g/kg, i.p.) and DCZ (10 μg/kg, i.p.) (ipGTT). Co-injected obese α-GsD mice displayed significantly improved glucose tolerance, as compared to obese control littermates (Fig. 2e). As observed with lean mice (Fig. 2b), DCZ-induced activation of α-cell Gs signaling resulted in a striking increase in plasma glucagon levels in obese α-GsD mice, but not in obese control littermates (Fig. 2f). Plasma

insulin levels were also significantly elevated in obese α-GsD mice co-injected with glucose and DCZ (GSIS) (Fig. 2g). Co-injection of obese α-GsD mice and their control littermates with insulin (1 U/kg, i.p.) and DCZ (10 μg/kg, i.p.) (ITT) did not reveal any differences in insulin sensitivity between the two groups of mice (Fig. 2h). These data indicate that acute activation of α-cell Gs signaling results in an insulinotropic effect that improves glucose homeostasis in both lean and obese, glucose-intolerant mice.

### Key role of α-cell adenosine A₂ₐ receptors in regulating α-cell function

Our next goal was to identify Gs-coupled GPCRs that are endogenously expressed by pancreatic α-cells with relatively high selectivity. Analysis of previously published scRNAseq data from human and mouse islets[31,32] led to the identification of three Gs-coupled receptors that are selectively expressed in α-cells, as compared to other islet cell types. These receptors include GPR119 (gene name: *Gpr119*), the A₂ₐ adenosine receptor (A2AR; gene name: *Adora2a*), and the β₁-adrenergic receptor (β₁-AR, gene name: *Adrb1*) (Supplementary Fig. 3a, b). Because of the availability of highly selective A2AR agonists and antagonists[39] and floxed A2AR mice[40], we decided to explore the potential metabolic roles of α-cell A2ARs. This receptor subtype is also expressed at low to moderate levels in mouse islet δ-cells (Supplementary Fig. 3a; Fig. 4a).

A previous study using an enzyme-coated electrode biosensor demonstrated that the extracellular levels of adenosine in rodent islets are inversely correlated with glucose levels in the surrounding medium[41]. We therefore speculated that adenosine-mediated activation of α-cell A2ARs might play an important role in promoting glucagon release when glucose levels are low. To test this hypothesis, we perifused islets from WT mice with a selective A2AR agonist, UK 432097 (50 nM). As expected for an agonist acting on an α-cell Gs-coupled receptor, UK 432097 treatment of WT islets resulted in significant increases in glucagon release at both low and high glucose levels (G3 and G12, respectively), resembling the pattern observed with DCZ-treated α-GsD islets (Fig. 3a, b). This UK 432097 effect was completely abolished by pretreatment of WT islets with SCH 442416 (0.5 μM), a selective A2AR antagonist, confirming the involvement of A2ARs (Fig. 3a, b). Strikingly, at low, but not at high glucose concentrations, application of SCH 442416 alone caused a pronounced decrease in glucagon release (Fig. 3a), suggesting that the α-cell A2AR signaling is required to maintain sufficient glucagon release under hypoglycemic conditions.

A2AR-mediated activation of Gs is predicted to increase intracellular cAMP levels via Gs-induced activation of adenylyl cyclase. To monitor A2AR-stimulated cAMP accumulation in α-cells, we employed islets from α-CAMPER mice that express a cAMP biosensor exclusively in α-cells[36]. We found that A2AR agonist treatment (UK 432097, 5 or 20 nM) of α-CAMPER islets resulted in a small reduction in cAMP levels at G3 but caused a significant increase in cAMP accumulation at G12 (Fig. 3c, d). The A2AR agonist-induced decrease in cAMP levels at G3 is probably due to the facts that A2ARs are already strongly stimulated by high endogenous adenosine levels[41] and that α-cell cAMP levels are already high in a low glucose environment[42]. In agreement with these observations, addition of the A2AR antagonist SCH 442416 (0.5 μM)

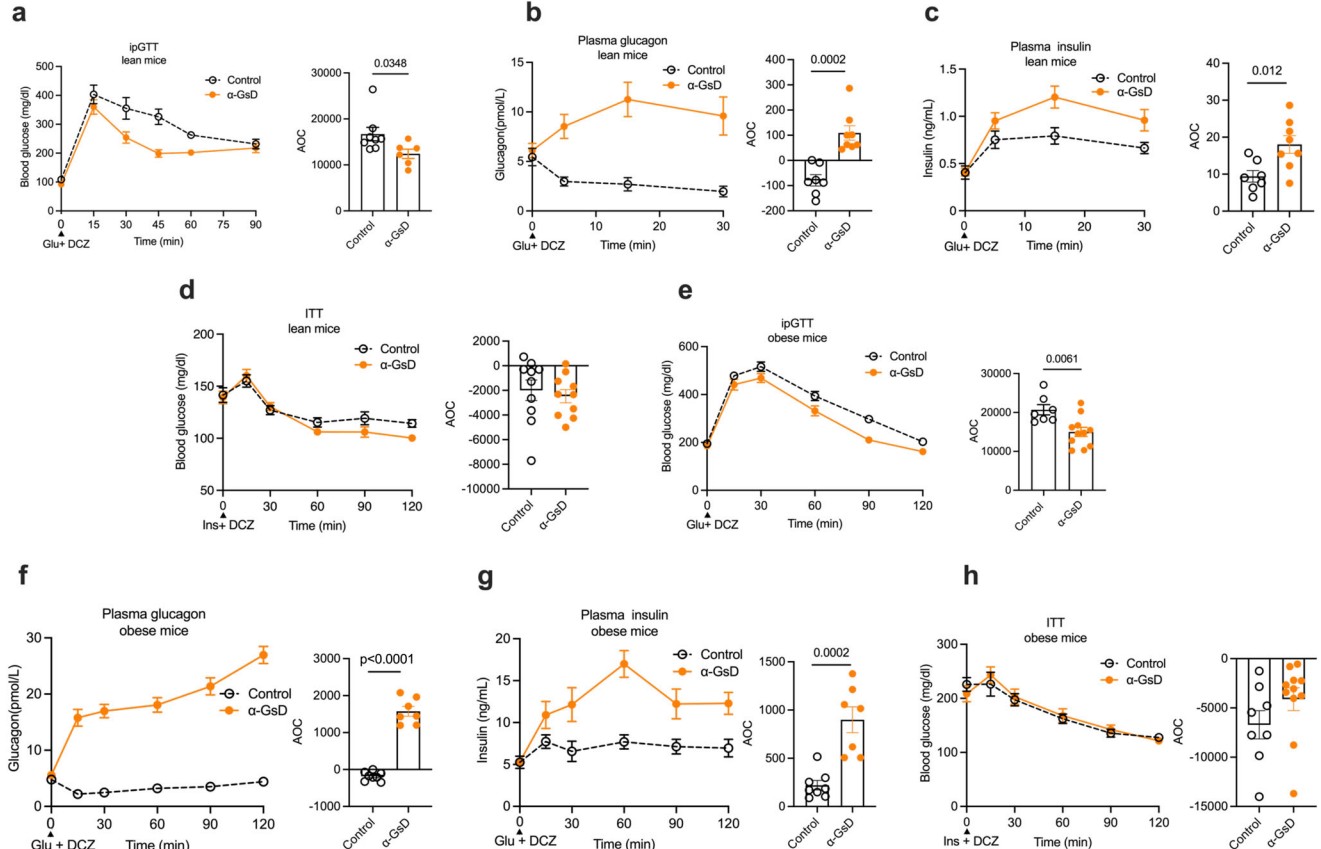

**Fig. 2 | Acute activation of the α-cell GsD designer receptor improves glucose tolerance in both lean and obese mice.** α-GsD mice and control littermates consuming regular chow (lean mice) or a high-fat diet (HFD; obese mice) were subjected to a series of metabolic tests. **a** Glucose tolerance test (ipGTT). Lean mice that had been fasted overnight were co-injected (i.p.) with glucose (2 g/kg) and DCZ (10 µg/kg) (control, $n = 8$; α-GsKO, $n = 6$). Changes in plasma glucagon (**b**) and plasma insulin (**c**) levels following i.p. co-injection of lean mice with glucose and DCZ (control, $n = 7$; α-GsKO, $n = 8$). **d** Insulin tolerance test (ITT). Lean mice that had been fasted for 4 h after were injected (i.p.) with a mixture of insulin (0.75 U/kg) and DCZ (control, $n = 10$; α-GsKO, $n = 10$). **e** ipGTT. Obese mice that had been fasted overnight were co-injected (i.p.) with glucose (1 g/kg) and DCZ (10 µg/kg) (control, $n = 7$; α-GsKO, $n = 11$). Changes in plasma glucagon (**f**) and plasma insulin (**g**) levels following i.p. co-injection of obese mice with glucose and DCZ (control, $n = 8$; α-GsKO, $n = 7$). **h** Insulin tolerance test (ITT). Following a 4 h fast, obese mice were injected (i.p.) with a mixture of insulin (1 U/kg) and DCZ (10 µg/kg) (control, $n = 8$; α-GsKO, $n = 11$). Blood samples were collected from the tail vein at the indicated time points. All experiments were carried out with male littermates that were at least 14 weeks old. Obese mice consumed the HFD for at least 8 weeks. Data are given as means ± SEM. Data were subjected to two-tailed Student's $t$ test (AOC bars) or to two-way repeated measures ANOVA for time with Bonferroni post hoc test for comparison of individual time points (**a**–**c**, **e**–**g**). AOC, area over the curve. Numbers in the bar graphs or next to specific data points data points refer to $p$ values. AOC, area of the curve. Source data are provided as a Source Data file.

led to a very robust reduction of cAMP levels at G3 (Fig. 3c), raising the possibility that the inhibitory effect of the A2AR agonist at G3 was caused by A2AR desensitization or the activation of other, yet unknown, signaling pathways that interfere with adenosine-induced cAMP production. On the other hand, at G12, α-cell A2AR signaling is predicted to be reduced due to low extracellular adenosine levels[41], explaining why the A2AR agonist (UK 432097) promoted cAMP production and the A2AR antagonist (SCH 442416) caused only a minor reduction in cAMP levels, as compared to basal levels prior to the addition of ligands (Fig. 3d).

To confirm the involvement of intraislet adenosine in stimulating α-cell A2ARs at low glucose levels (G3), we treated control islets with adenosine deaminase (ADA, 5 U/ml) which leads to the conversion of adenosine to inosine, a metabolite that is unable to activate A2ARs[43,44]. As shown in Fig. 3e, ADA treatment of control islets resulted in a pronounced decrease in glucagon secretion at G3. Strikingly, this effect was absent in islets prepared from mice that selectively lacked A2ARs receptors or the α-subunit of $G_s$ ($G\alpha_s$) in α-cells (α-A2A-KO mice and α-GsKO mice, respectively; see below) (Fig. 3e). Moreover, basal glucagon secretion at G3 was drastically reduced in α-A2A-KO islets (Fig. 3e). Taken together, these data strongly suggest that α-cell A2AR/

$G_s$ signaling plays a key role in stimulating sufficient glucagon release under hypoglycemic conditions.

Previous studies have shown that increases in intracellular $Ca^{2+}$ levels resulting from the activation of α-cell $G_q$-coupled receptors can also trigger glucagon release from α-cells[35,45,46]. Prompted by this finding, we also studied islets from α-GCaMP6s mice (α-GCaMP6s islets) that express a $Ca^{2+}$ reporter exclusively in α-cells[36]. Treatment of α-GCaMP6s islets with UK 432097 (A2AR agonist) or SCH 442416 (A2AR antagonist) had no significant effect on intracellular $Ca^{2+}$ levels at G3 or G12 (Supplementary Fig. 3c, d), confirming that G proteins of the $G_q$ family do not contribute to α-cell A2AR-mediated glucagon secretion.

## Stimulation of α-cell A2ARs also promotes glucagon release in human islets

We next explored whether the functional role of the α-cell A2AR signaling cascade observed with mouse islets was also operative in human islets. To address this question, we treated islets obtained from human donors with the A2AR agonist UK 432097 (5 nM). As observed with mouse islets, UK 432097 treatment of human islets resulted in significant increases in glucagon secretion at both low and

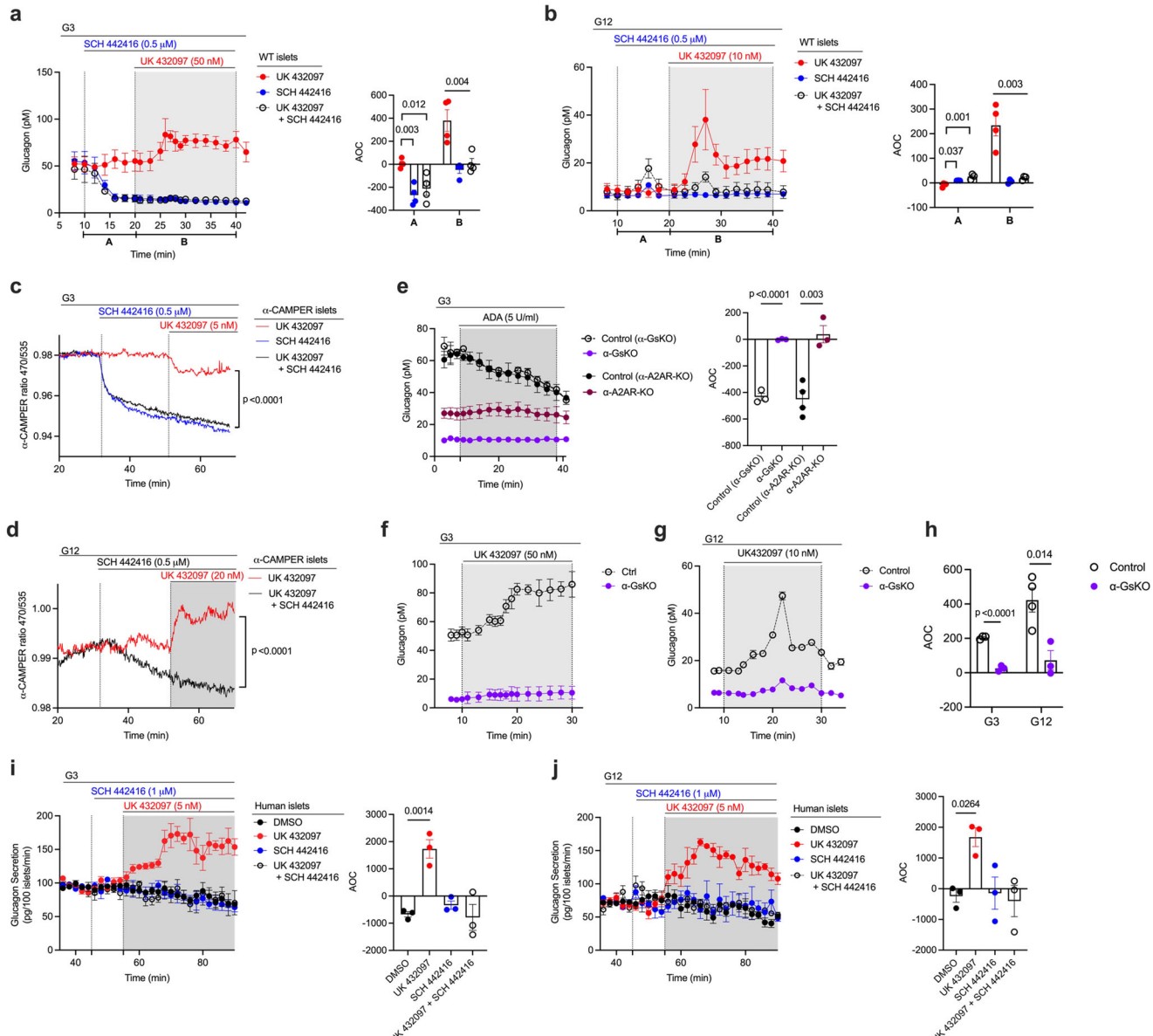

**Fig. 3 | Activation of α-cell A2ARs strongly stimulates glucagon secretion from mouse and human islets. a**, **b** Measurement of glucagon secretion from perifused mouse WT islets. Experiments were carried out at low and high glucose levels (3 mM [G3] and 12 mM [G12], respectively) either in the presence of UK432097 (A2AR-selective agonist: 50 nM at G3, 10 nM at G12) or SCH442416 (A2AR-selective antagonist, 0.5 µM) alone or in the presence of both ligands (*n* = 4 mice per group). cAMP production in α-cells from α-CAMPER mice at G3 (**c**) or G12 (**d**) in the presence of UK432097 (100 nM) or SCH442416 (0.5 µM) or in the presence of both ligands. **e**–**h** Glucagon release studies with islets lacking Gα$_s$ or A2ARs in their α-cells. Islets were prepared from α-GsKO and α-A2AR-KO mice and their corresponding littermates. In (**e**), islets were treated with ADA (5 U/ml) at G3 to enzymatically remove extracellular adenosine (*n* = 3 or 4 mice per group). In (**f**, **g**), α-GsKO and control islets were treated with UK432097 at G3 and G12 (*n* = 3 or 4 mice

per group). AOC values (**h**) for glucagon release data shown in (**f**) and (**g**) (time period: 8–30 min). **i**, **j** A2AR activation stimulates glucagon secretion from human islets. Islets from human donors were perifused with G3 (**i**) or G12 (**j**), respectively, either in the presence of vehicle (DMSO), UK432097 or SCH442416 alone, or in the presence of both UK432097 and SCH442416 (*n* = 3 donors per group). Islets were obtained from male or female mice that were 14–24 weeks old. AOC values were calculated for different stimulation periods. Data are shown as means ± SEM (3 or 4 independent perifusions with 75–100 islets per perifusion chamber). Data were analyzed via two-tailed Student's *t* test (AOC values in **a**, **b**, **e**, **h**–**j**) or two-way repeated measures ANOVA with time (**c**, **d**). Numbers above horizontal lines in the bar graphs represent p values. ADA, adenosine deaminase. AOC, area of the curve. Source data are provided as a Source Data file.

high glucose concentrations (Fig. 3i, j). This response was abolished in the presence of the A2AR antagonist SCH 442416 (1 µM) (Fig. 3i, j), indicative of the involvement of A2ARs. However, while SCH 442416 treatment of WT mouse islets caused a pronounced decrease of glucagon release at G3 (Fig. 3a), we did not observe this effect in human islets (Fig. 3i). One possible explanation for this observation is that mouse islets were freshly prepared for glucagon release studies, whereas human islets were first cultured for several days under

conditions (see Methods for details) predicted to lower extracellular adenosine levels[41].

### α-Cell A2ARs stimulate glucagon secretion in vitro and in vivo

To elucidate the potential physiological relevance of α-cell A2ARs, we generated mice that lacked A2ARs selectively in α-cells. By crossing *Adora2a*$^{fl/fl}$ mice[40] with *Gcg-Cre*$^{ERT2}$ mice[34], we obtained *Adora2a*$^{fl/fl}$ *Gcg-Cre*$^{ERT2}$ mice. TMX treatment of *Adora2a*$^{fl/fl}$ *Gcg-Cre*$^{ERT2}$ mice resulted in a

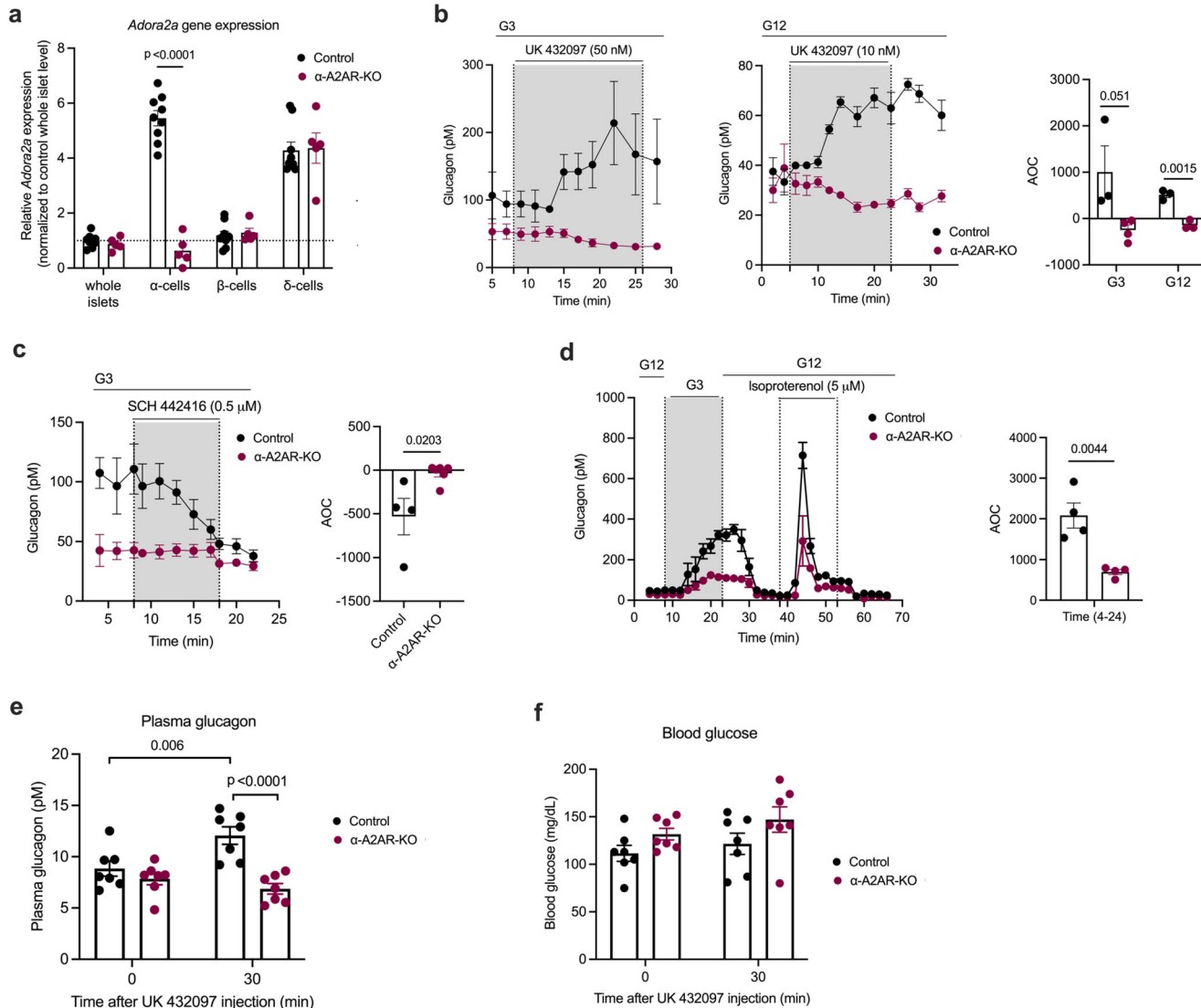

**Fig. 4 | Activation of α-cell A2ARs in vivo induces glucagon secretion. a** Absence of *Adora2a* transcript in α-cells of α-A2AR-KO mice. qRT-PCR analysis of *Adora2a* gene expression (encoded protein: A2AR) in whole islets, α-, β- and δ-cells. The different cell types were isolated via FACS sorting using islets from control and α-A2AR-KO mice (males). **b** Glucagon secretion from control and α-A2AR-KO islets treated with the A2AR-selective agonist UK432097 (50 nM at G3, 10 nM at G12). **c** Glucagon secretion from perifused control and α-A2AR-KO islets treated with the A2AR-selective antagonist SCH442416 (0.5 μM) at G3. **d** Glucagon release from perifused control and α-A2AR-KO islets in the presence of different glucose concentrations and isoproterenol (5 μM), a β-adrenergic receptor agonist. Plasma

glucagon (**e**) and blood glucose (**f**) levels measured after i.p. injection of control and α-A2AR-KO mice with UK432097 (5 mg/kg). AOC values were calculated for different stimulation periods. All experiments were carried out with male littermates (12–20 weeks old). Data are given as means ± SEM (in vivo studies: ($n$ = 7 per group); in vitro perifusion studies: 3–5 independent perifusions with 75–100 islets per perifusion chamber). Data were analyzed via two-tailed Student's $t$ test ((**a**) and AOC values in (**b**–**d**)) or two-way repeated measures ANOVA for time with Bonferroni post hoc test for comparison of time (**e**, **f**). Numbers in the AOC panels represent $p$ values. AOC, area of the curve. Source data are provided as a Source Data file.

robust reduction of *Adora2a* mRNA selectively in pancreatic α-cells (Fig. 4a). In the following, we refer to these mice simply as α-A2AR-KO mice. TMX-treated *Adora2a*[fl/fl] littermates that lacked the *Cre* transgene served as control animals in all studies in which α-A2AR-KO mice were used. The lack of α-cell A2ARs had no significant effect on pancreas weight, total pancreatic or islet glucagon and insulin content, or the expression of key α-cell genes including *Gcg* (Supplementary Fig. 4a–f).

Initially, we performed perifusion experiments using islets prepared from α-A2AR-KO mice and control littermates. Treatment of control islets with the UK 432097 A2AR agonist (50 nM at G3; 10 nM at G12) resulted in significant increases in glucagon secretion at both G3 and G12 (Fig. 4b). These effects were completely absent in α-A2AR-KO islets (Fig. 4b), confirming the involvement of α-cell A2ARs. We also

noted that basal glucagon secretion at G3 was significantly decreased ($P$ = 0.042861) in α-A2AR-KO islets (Fig. 4b, c). Consistent with studies carried out with WT islets (Fig. 3a), treatment of control islets with the SCH 442416 A2AR antagonist (0.5 μM) reduced glucagon release at G3 to levels observed with α-A2AR-KO islets (Fig. 4c). In contrast, SCH 442416 had no significant effect on basal glucagon secretion observed with A2AR-KO islets (Fig. 4c). G3-induced glucagon secretion (after exposure to G12) was also significantly reduced in α-A2AR-KO islets, as compared to control islets (Fig. 4d). Taken together, these in vitro data further corroborate the concept that α-cell A2ARs play a key role in maintaining adequate glucagon secretion when glucose levels are low.

In vivo studies showed that α-A2AR-KO mice maintained on regular chow did not differ from their control littermates in body

weight (Supplementary Fig. 4g), fed and fasting plasma glucagon and insulin concentrations (Supplementary Fig. 4h, i), and blood glucose levels (Supplementary Fig. 4j). We first injected α-A2AR-KO mice and control littermates with the A2AR agonist UK 43209 (5 mg/kg, i.p.). Strikingly, the UK 432097-induced increase in plasma glucagon levels observed with control mice was completely absent in α-A2AR-KO mice (Fig. 4e), indicating that activation of α-cell A2ARs triggers glucagon release in vivo. UK 432097 treatment also led to enhanced plasma insulin levels in all control mice, but this response was absent in α-A2AR-KO mice (except for one mouse) (Supplementary Fig. 4k). Blood glucose levels remained unaltered in UK 432097-injected control mice (Fig. 4f), most likely due to hyperglucagonemia-activated counterregulatory responses in the UK 432097-injected control mice.

We next subjected α-A2AR-KO mice and control littermates that had been maintained on an obesogenic HFD for at least 8 weeks to a series of metabolic tests. Both groups of mice showed similar weight gain, glucose tolerance (ipGTT), and insulin sensitivity (ITT) (Supplementary Fig. 4l–n), suggesting that α-cell A2AR deficiency has no detectable effect on glucose homeostasis in obese, glucose-intolerant mice.

Since glucagon plays a crucial role as a counter-regulatory hormone in response to hypoglycemia in lean mice[45], we investigated whether α-cell A2ARs contribute to the counter-regulatory increase in plasma glucagon levels under hypoglycemic conditions. To induce hypoglycemia, we injected α-A2AR-KO and control mice with insulin (1 U/kg, i.p.). During insulin-induced hypoglycemia, plasma glucagon levels and blood glucose levels did not differ significantly between the two groups of mice (Supplementary Fig. 4o, p).

2-Deoxy-D-glucose (2-DG) is a glucose analog that interferes with the production of glucose-6-phosphate from glucose, leading to glucopenia in the brain and other tissues[47–50]. As a result, 2-DG-induced glucopenia can activate α-cell signaling pathways that stimulate glucagon secretion[35,46]. We found that 2-DG-induced increases in plasma glucagon, plasma insulin, and blood glucose levels were not significantly affected by the lack of α-cell A2ARs (Supplementary Fig. 4q–s).

### Selective inactivation of the gene coding for Gα<sub>s</sub> (*Gnas*) in mouse α-cells

To further explore the metabolic role of α-cell G<sub>s</sub> (gene name: *Gnas*), we developed mice in which we selectively inactivated the *Gnas* gene in pancreatic α-cells in adult mice. To generate this new mouse line, we crossed *Gnas^{fl/fl}* mice[51] with *Gcg-Cre^{ERT2}* transgenic mice[34] to obtain *Gnas^{fl/fl} Gcg-Cre^{ERT2}* mice. TMX treatment of adult *Gnas^{fl/fl} Gcg-Cre^{ERT2}* mice caused a robust reduction of *Gnas* mRNA levels specifically in pancreatic α-cells, as compared to TMX-injected *Gnas^{fl/fl}* mice lacking the *Cre* transgene (control littermates) (Fig. 5a). Disruption of the *Gnas* gene in α-cells of TMX-treated *Gnas^{fl/fl} Gcg-Cre^{ERT2}* mice (referred to simply as α-GsKO mice hereafter) had no significant effect on *Gnaq* expression levels (Fig. 5b). *Gnaq* codes for the α-subunit of G<sub>q</sub> which, after activation by GPCRs, can stimulate glucagon release from pancreatic α-cells[35,46,52,53].

Immunofluorescence studies with pancreatic sections from α-GsKO mice confirmed the absence of Gα<sub>s</sub> protein in glucagon-expressing α-cells (Fig. 5c). Gα<sub>s</sub> expression remained unaffected in β-cells or other non-α-cells in α-GsKO islets (Fig. 5c). The selective deletion of Gα<sub>s</sub> in pancreatic α-cells had no significant effect on body weight, pancreas weight, or α-cell and β-cell mass (Supplementary Fig. 5a–e).

### In vitro studies with α-GsKO islets and pancreata

To provide functional evidence for the absence of α-cell G<sub>s</sub> signaling in α-GsKO islets, we carried out glucagon secretion studies with perfused pancreatic islets prepared from α-GsKO mice and control littermates. Specifically, we treated islets with isoproterenol which activates G<sub>s</sub>-coupled β-adrenergic receptors which are known to be expressed by α-cells[54]. Isoproterenol treatment led to a robust stimulation of glucagon release in control islets but showed only residual activity in α-GsKO islets at either low or high glucose levels (G3 and G12, respectively) (Fig. 5d, e,), indicating that α-cell G<sub>s</sub> signaling is disrupted in α-GsKO islets. Interestingly, in the absence of isoproterenol (or other ligands), glucagon release at both G3 and G12 was greatly reduced in α-GsKO islets, as compared to control islets (Fig. 5d, e).

As discussed above, glucagon secretion studies with α-A2AR-KO islets indicated that G<sub>s</sub> signaling contributes to the intrinsic control of glucagon secretion in α-cells at low glucose. Consistent with this concept, G3-induced glucagon secretion (following exposure to G12) was significantly reduced in α-GsKO islets, as compared to control islets (Fig. 5f). We also treated α-GsKO and control islets with a vasopressin 1b receptor (V1bR) agonist (d[Leu⁴,Lys⁸]VP, 10 nM). Recent work demonstrated that arginine vasopressin stimulates glucagon secretion via activation of G<sub>q</sub>-coupled V1bRs expressed by pancreatic α-cells[35,46]. Despite the absence of α-cell Gα<sub>s</sub>, the V1bR agonist was able to stimulate glucagon release. However, the amount of glucagon released after agonist activation of α-cell V1bRs was significantly reduced in α-GsKO islets (Fig. 5f).

Previous studies have shown that somatostatin (SST) release from δ-cells inhibits glucagon secretion from α-cells (reviewed in ref. 55). To investigate whether this effect was altered in α-GsKO islets, we incubated control and α-GsKO islets with a combination of two SST receptor inhibitors (PRL2915 + PRL3195, 1 μM each) which are known to block the major SST receptor subtypes expressed by mouse α-cells (SSTR2, SSTR3, and SSTR5)[56]. We monitored both basal glucagon and isoproterenol-induced glucagon secretion. The dramatic reductions in basal and isoproterenol-induced glucagon secretion caused by α-cell G<sub>s</sub> deficiency in the absence of SST receptor blockade persisted in the presence of the SST receptor antagonists (Supplementary Fig. 5f, g). These data suggest that it is unlikely that altered SST release from δ-cells contributes to the functional deficits displayed by α-GsKO mice or α-GsKO islets.

### α-GsKO mice show reduced plasma glucagon levels in vivo

We next subjected α-GsKO mice and control littermates maintained on regular chow to several in vivo metabolic tests. Interestingly, under fasting conditions, α-GsKO mice displayed a significant reduction in plasma glucagon levels, as compared to fasted control littermates (Fig. 6a). This effect was not observed in mice that had free access to food (Fig. 6a). Plasma insulin and blood glucose levels did not differ significantly between α-GsKO and control mice under both fed and fasting conditions (Fig. 6b, c). Additional metabolic tests showed that α-GsKO and control mice displayed no significant differences in glucose and insulin tolerance (Supplementary Fig. 6a, b).

To further corroborate the involvement of α-cell G<sub>s</sub> signaling in A2AR-stimulated glucagon secretion, we treated control islets and α-GsKO islets with the A2AR agonist UK 432097 (50 nM) (Fig. 3f, g). We found that UK 432097-induced glucagon secretion was absent or nearly completely abolished in α-GsKO islets at both G3 and G12, respectively. In addition, we carried out vivo experiments with PSB 0777 (1 mg/kg, i.p.), another selective A2AR agonist[57]. In agreement with the in vitro data (Fig. 3f, g), PSB 0777-induced increases in plasma glucagon levels were abolished in α-GsKO mice, as compared to control littermates (Supplementary Fig. 6c). Plasma insulin and blood glucose levels were similar in PSB 0777-treated in both groups of mice (Supplementary Fig. 6d, e). These data clearly indicate that A2AR-stimulated glucagon secretion requires α-cell G<sub>s</sub> signaling.

When maintained on a HFD for 8 weeks, both α-GsKO mice and control littermates showed similar weight gain (Supplementary

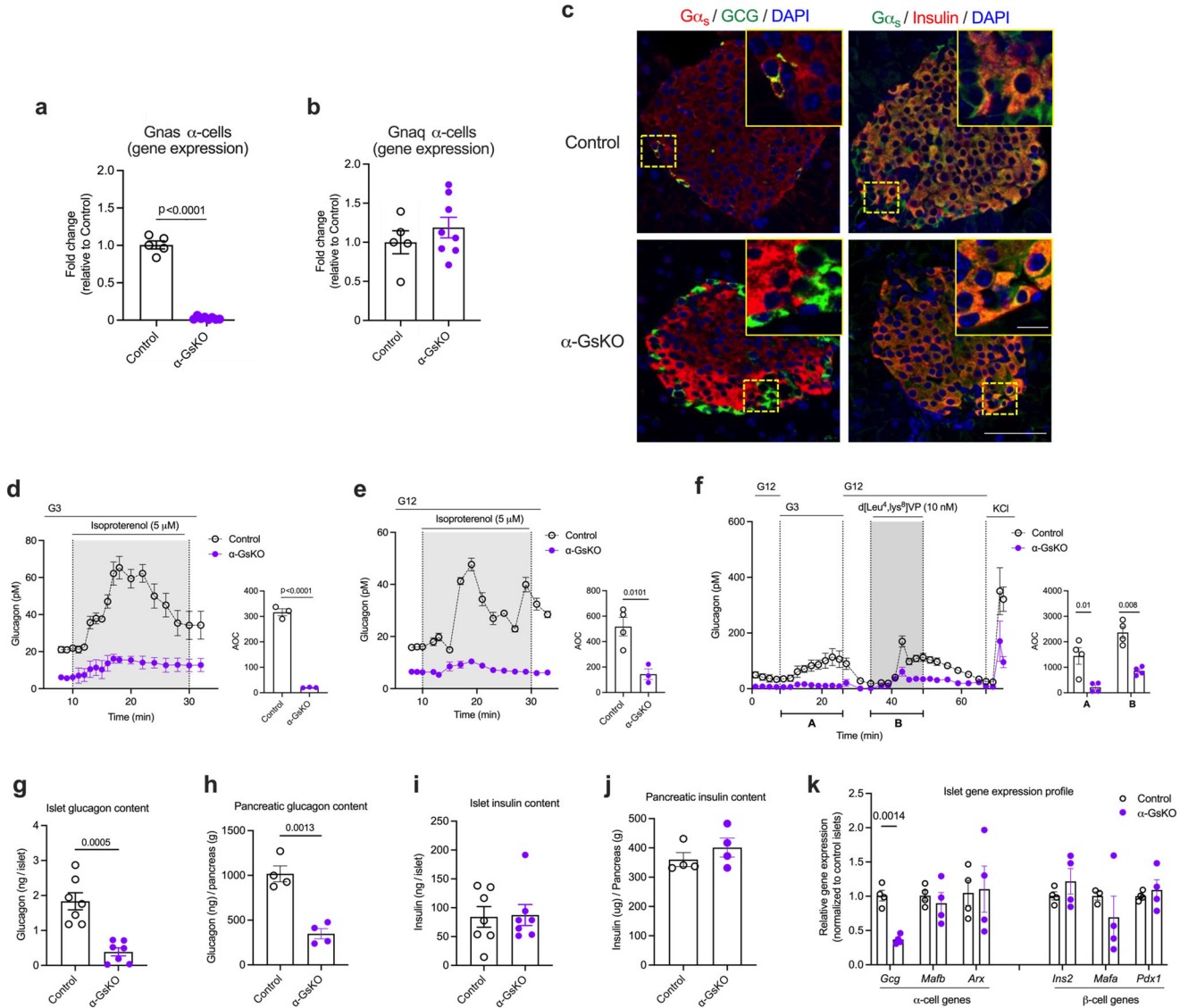

**Fig. 5 | Selective lack of Gα$_s$ in α-cells leads to impaired glucagon secretion and reduced islet glucagon content and *Gcg* expression. a–c** Data confirming the absence of *Gnas* mRNA (encoded protein: Gα$_s$) or Gα$_s$ protein in α-cells of α-GsKO mice (α-cell *Gnas$^{-/-}$* mice). **a** Absence of *Gnas* mRNA in α-cells isolated from α-cell *Gnas$^{-/-}$* mice via FACS (α-GsKO, $n = 8$ mice; control, $n = 5$ mice). Gene expression data were obtained via qRT-PCR. **b** The lack of α-cell *Gnas* expression does not affect α-cell *Gnaq* expression (encoded protein: Gα$_q$) (α-GsKO, $n = 8$ mice; control, $n = 5$ mice). **c** Immunofluorescence staining of pancreatic slices from α-GsKO mice and control littermates. Slices were co-stained with either an anti-Gα$_s$ antibody (Alexa Fluor, red) and an anti-glucagon antibody (Alexa Fluor, green), or an anti-Gα$_s$ antibody (Alexa Fluor, green) and an anti-insulin antibody (Alexa Fluor, red), respectively. The inserts in the upper right of each panel show enlarged islets areas. Scale bar in inserts: 10 μm; scale bars in non-enlarged images: 50 μm. Contrast was adjusted for improved visualization. **d, e** Glucagon secretion studies carried out with perifused islets prepared from α-GsKO mice and control littermates. While treatment of control islets with 5 μM isoproterenol (β-adrenergic receptor agonist) strongly stimulated glucagon release at both G3 and G12, this response was almost

completely abolished in α-GsKO islets ($n = 3$ or 4 mice per group). **f** Stimulation of glucagon release by low glucose (G3), a V1b receptor agonist (d[Leu$^4$, Lys$^8$]VP, 10 nM), and KCl (30 mM) from perifused α-GsKO mice and control islets ($n = 4$ mice per group). Glucagon content of islets and pancreata from control and α-GsKO mice (**g, h**: $n = 7$ and $n = 4$ mice per group, respectively). Insulin content of islets and pancreata from control and α-GsKO mice (**i, j**: $n = 7$ and $n = 4$ mice per group, respectively). **k** Expression levels of key α- and β-cell genes determined with RNA prepared from control and α-GsKO islets ($n = 4$ mice per group). AOC values were calculated for different stimulation periods. The data shown in (**g–k**) were generated using islets obtained from male mice (age: ~30-weeks). For islet perifusion studies, 75–100 islets per chamber were used. Islets were prepared from male mice (age: 16–20 weeks). Immunofluorescence images are representative of three independent experiments. Data are given as means ± SEM. Data were analyzed via two-tailed Student's *t* test (**a, g, h, k**, and AOC values in **d–f**). Numbers in the AOC panels represent *p* values. AOC, area of the curve. Source data are provided as a Source Data file.

Fig. 7a). Notably, obese α-GsKO mice showed decreased plasma glucagon levels under both fed and fasting conditions, as compared to obese control littermates (Fig. 6d). Despite reduced plasma glucagon levels, obese α-GsKO and control mice did not differ significantly in fed and fasting plasma insulin and blood glucose levels (Fig. 6e, f), or glucose tolerance and insulin sensitivity (Supplementary Fig. 7b, c).

## Glucagon mRNA and protein are greatly reduced in islets lacking functional Gα$_s$

We speculated that the hypoglucagonemia phenotype displayed by the α-GsKO mice (Fig. 6a, d) was due to reduced basal glucagon release from α-GsKO islets (Fig. 5d–f). To further explore this hypothesis, we measured the amount of glucagon present in pancreatic islets and whole pancreata isolated from control and α-GsKO mice. Strikingly,

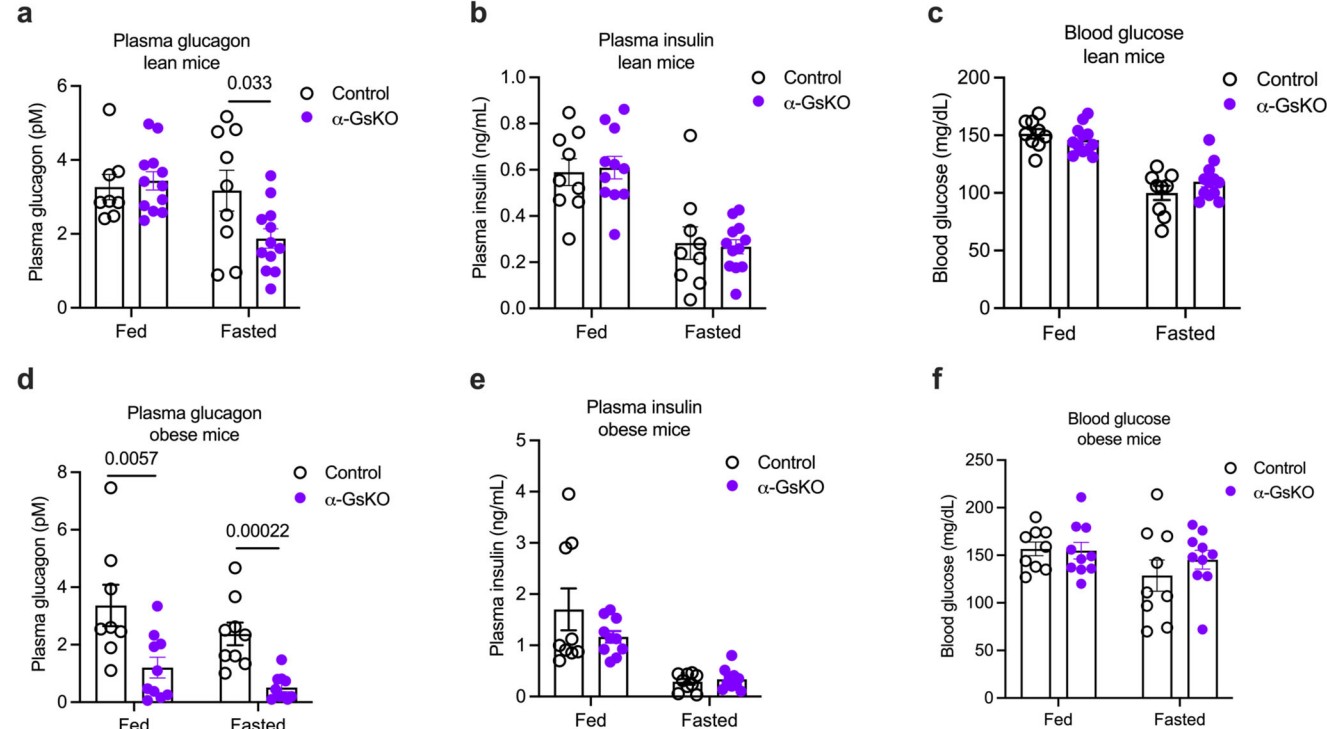

**Fig. 6 | Disruption of α-cell Gₛ signaling results in reduced plasma glucagon levels.** Plasma hormone ((**a**), glucagon; (**b**), insulin) and blood glucose (**c**) levels in α-GsKO mice and control littermates consuming regular chow (lean mice). Mice had either free access to food (fed) or were fasted overnight. Plasma hormone ((**d**), glucagon; (**e**), insulin) and blood glucose (**f**) levels in α-GsKO mice and control littermates maintained on a HFD (obese mice). Mice had either free access to food (fed) or were fasted overnight. Blood samples were collected from the tail vein. All experiments were carried out with male littermates. At the time of testing, lean mice were 14 weeks old. Obese mice were maintained on the HFD for at least 8 weeks, after having consume regular chow for 18 weeks. Data are given as means ± SEM (lean mice: control, $n = 9$; α-GsKO, $n = 12$; obese mice: control, $n = 9$; α-GsKO, $n = 10$). Data were analyzed via two-tailed Student's $t$ test (**a, d**). Numbers above the horizontal lines in the bar graphs represent $p$ values. Source data are provided as a Source Data file.

glucagon content was significantly reduced in α-GsKO islets and pancreata, as compared to the corresponding control preparations (Fig. 5g, h). Pancreatic SST content was similar in control and α-GsKO mice (Supplementary Fig. 7d). Moreover, islet and pancreatic insulin content did not differ significantly between the two groups of mice (Fig. 5i, j). Similar findings were obtained with pancreata prepared from obese α-GsKO and control mice (Supplementary Fig. 7e, f).

qRT-PCR studies demonstrated that the expression of the *Gcg* gene was significantly decreased in α-GsKO islets, as compared with control islets (Fig. 5k). The expression levels of other key α-cell (*Mafb* and *Arx*) and β-cell (*Ins2, Mafa,* and *Pdx1*) genes were not significantly affected by α-cell Gαₛ deficiency (Fig. 5k). To investigate the mechanism underlying the decrease in *Gcg* expression caused by α-cell Gₛ deficiency, we carried out additional studies with cultured mouse α-TC6 cells, an adenoma-derived clonal α-cell line. Treatment of α-TC6 cells with PKI 14-22 (10 μM), a highly selective inhibitor of PKA, a protein kinase activated by Gₛ signaling, led to significantly reduced *Gcg* RNA levels (Supplementary Fig. 7g), in agreement with published work demonstrating that activation of PKA promotes *Gcg* transcription[58,59].

In sum, these data strongly suggest that disruption of α-cell Gₛ signaling suppresses the expression of the *Gcg* gene, leading to reduced glucagon synthesis and storage. This deficit is most likely responsible for the hypoglucagonemia phenotype displayed by the α-GsKO mice (Fig. 6a, d).

## α-Cell Gₛ contributes to glucagon secretion caused by hypoglycemia and glucopenia

To explore the potential involvement of α-cell Gₛ signaling in hypoglycemia-induced glucagon secretion, we treated control and α-GsKO mice with exogenous insulin (1 U/kg, i.p.) (Fig. 7a, b). Strikingly, insulin-induced hypoglycemia resulted in significantly smaller elevations of plasma glucagon levels in α-GsKO mice (Fig. 7a), probably partially due to the reduction in pancreatic glucagon content caused by α-cell Gαₛ deficiency (Fig. 5g, h). However, this deficit had no significant effect on the magnitude of insulin-induced decreases in blood levels (Fig. 7b). One possible explanation for this observation is that the mice received a supraphysiological dose of insulin that masked the metabolic effects of altered plasma glucagon levels.

We next examined whether α-cell Gαₛ deficiency affected the increase in plasma glucagon levels caused by glucopenic conditions (2-DG test). Interestingly, glucagon secretion in response to 2-DG injection (500 mg/kg, i.p.) was significantly impaired in α-GsKO mice (Fig. 7c). It is likely that reduced pancreatic glucagon content (Fig. 5g, h) contributed to this deficit. At the same time, the 2-DG-dependent increase in glucose-driven insulin secretion was also significantly reduced (Supplementary Fig. 8a, b). In α-GsKO mice, elevated plasma insulin levels returned to baseline more rapidly than in control littermates (Supplementary Fig. 8a, b). It is likely that this effect is responsible for the elevated blood glucose levels observed with 2-DG-treated α-GsKO mice, as compared to 2-DG-treated control littermates (Fig. 7d).

## α-Cell Gₛ signaling is essential for GIP- and GIP/alanine-induced glucagon secretion

A recent study demonstrated that alanine and GIP have a synergistic effect on glucagon secretion, and that this activity is crucial for the proper regulation of postprandial insulin secretion[36]. El et al.[36] also showed that the combined treatment of mice with GIP and alanine

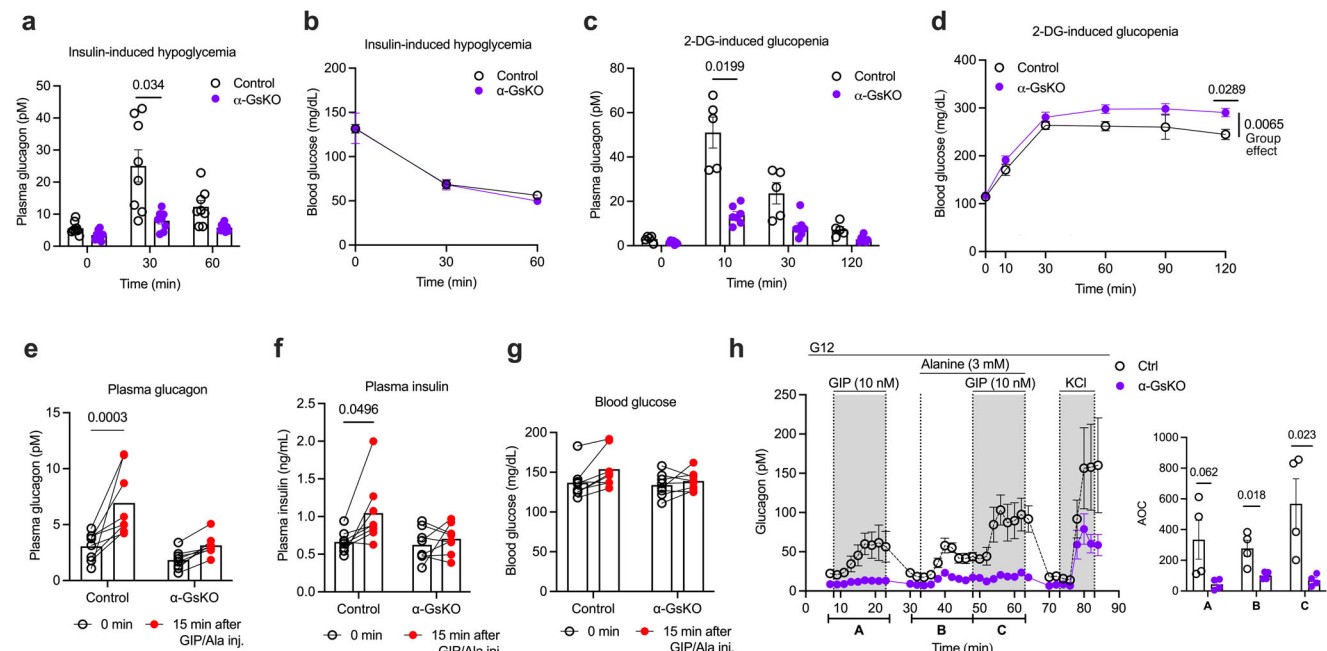

**Fig. 7 | Lack of α-cell Gₛ signaling leads to impaired glucagon release under different experimental conditions in vivo. a, b** Glucagon release following insulin-induced hypoglycemia. α-GsKO mice and control littermates were injected with insulin (1 U/kg, i.p.), and plasma glucagon (**a**) and blood glucose (**b**) levels were measured at the indicated time points (*n* = 8 per group). **c, d** Glucagon secretion after 2-DG-induced glucopenia. α-GsKO and control mice were injected with 2-DG (500 mg/kg, i.p.), and plasma glucagon (**c**) and blood glucose (**d**) levels were measured at the indicated time points (control, *n* = 5; α-GsKO, *n* = 7). **e–g** Treatment of mice with a mixture of GIP and alanine. α-GsKO and control mice were injected i.p. with a combination of GIP (4 nmol/kg) and alanine (0.325 g/kg), and plasma glucagon (**e**), plasma insulin (**f**), and blood glucose (**g**) levels were measured 15 min later (*n* = 8 per group). **h** Treatment of isolated islets with a GIP/alanine mixture to

induce glucagon secretion. Perifused islets isolated from control and α-GsKO mice were treated with a combination of GIP (10 nM) and alanine (3 mM) at G12. KCl (30 mM) was added at the end of the experiment (*n* = 4 mice per group). Note that GIP- and GIP/alanine-induced glucagon secretion was virtually abolished in α-GsKO islets. AOC values were calculated for different stimulation periods. All experiments were carried out with male littermates (14–20 weeks old). Blood was collected from the tail vein. Data are given as means ± SEM (in vitro studies: 4 perifusions with 75–100 islets per perifusion chamber). Data were analyzed via two-way repeated measures ANOVA for time with Bonferroni post hoc test for comparison of time (**a, c, d–f**) or two-tailed Student's *t* test (AOC values in (**h**)). Numbers above the horizontal lines in the bar graphs represent *p* values. AOC, area of the curve. Source data are provided as a Source Data file.

mimics the major metabolic effects of a mixed meal. Previous work has shown that the GIP receptor is selectively linked to Gₛ[60]. To investigate whether α-cell Gₛ signaling is involved in postprandial glucagon secretion, we treated control and α-GsKO mice with a combination of GIP (4 nmol/kg, i.p.) and alanine (0.325 g/kg, i.p.) (Fig. 7e–g). Strikingly, the combination of GIP and alanine synergistically increased plasma glucagon and insulin levels in control mice but not in α-GsKO littermates (Fig. 7e, f). Blood glucose levels remained unaffected by these changes in plasma hormone levels (Fig. 7g).

In agreement with the in vivo data (Fig. 7e) and the findings by ref. 36, treatment of isolated control islets with a GIP (10 nM)/alanine (3 mM) mixture led to a robust increase in glucagon secretion (Fig. 7h). To mimic the postprandial state, studies were carried out at elevated glucose levels (G12). This response was significantly greater than that observed after addition of GIP alone (Fig. 7h). Importantly, the ability of GIP and of the alanine/GIP mixture to stimulate glucagon secretion was completely abolished in α-GsKO islets (Fig. 7h). On the other hand, KCl (30 mM) treatment of α-GsKO islets resulted in a robust stimulation of glucagon secretion (Fig. 7h). However, the magnitude of this response was smaller than that observed with KCl-treated control islets (Fig. 7h), most likely due to the reduced glucagon content of α-GsKO islets (Fig. 5g, h).

## Discussion

Given the important physiological and pathophysiological roles of pancreatic α-cells, the identification of signaling proteins or pathways that regulate glucagon release from α-cells may lead to the identification of new classes of therapeutic targets. In this study, we used a

series of newly generated mutant mouse models to explore the potential in vivo metabolic roles of α-cell Gₛ signaling.

First, we generated and analyzed a mouse line that expressed a Gₛ-coupled DREADD (GsD) selectively in α-cells (α-GsD mice), enabling us to explore the in vivo metabolic consequences of selectively activating α-cell Gₛ signaling in vivo. We found that treatment of α-GsD mice, but not of control mice, with a selective GsD agonist (DCZ) led to robust increases in plasma glucagon and plasma insulin levels (Fig. 1d, e). Recent studies have shown that glucagon release from pancreatic α-cells can stimulate insulin release from adjacent β-cells in a paracrine fashion, in particular when glucose levels are high[11–15]. These recent findings provide a possible explanation for the observation that stimulation of α-cell Gₛ signaling in vivo leads to an increase of both plasma glucagon and insulin levels.

Moreover, DCZ treatment of α-GsD mice led to pronounced improvements in glucose tolerance in both lean and obese mice α-GsD mice (Fig. 2a, e). Most likely, this beneficial metabolic effect is due to elevated plasma insulin levels resulting from enhanced glucagon release. These data clearly indicate that α-cell Gₛ signaling can have beneficial metabolic effects on whole body glucose homeostasis. This observation is of considerable translational relevance, in particular since activation of α-cell Gₛ signaling significantly improved glucose tolerance in obese mice (Fig. 2e).

In agreement with the outcome of the in vivo studies, DCZ treatment of pancreatic islets prepared from α-GsD mice resulted in significant increases in glucagon release (Fig. 1g). We noted that basal glucagon release was markedly increased in α-GsD islets, as compared to control islets (Supplementary Fig. 1k). Previous studies have shown

                    

that the GsD designer receptor can signal, to a variable degree, in a ligand-independent fashion[25,33]. Thus, it is likely that the elevated glucagon levels observed with α-GsD islets in the absence of an activating ligand are most likely caused by the constitutive signaling via α-cell GsD. While this effect was easily detectable in α-GsD islets in vitro, basal plasma glucagon levels were similar in α-GsD mice and control littermates in vivo (Fig. 1a, d). This latter observation suggests that constitutive GsD signaling is unlikely to have a major impact on the outcome of the in vivo studies and that other physiological mechanisms, including, for example, circulating hormones and nutrients, paracrine factors, and neuronal pathways, limit constitutive GsD signaling in vivo.

Published scRNA-seq data[31,32] indicated that the $A_{2A}$ adenosine receptor (A2AR) is enriched in mouse and human pancreatic α-cells, as compared to other cell types present in pancreatic islets (Supplementary Fig. 3a, b). Since the A2AR is a prototypic $G_s$-coupled receptor[39], we decided to explore the potential metabolic roles of α-cell A2ARs in vitro and in vitro. Treatment of WT mouse islets with a selective A2AR agonist (UK 432097) resulted in a marked increase in glucagon release (Fig. 3a, b), in agreement with an earlier report[61]. Glucagon release studies with human islets confirmed that this cellular pathway is also operative in human α-cells (Fig. 3i, j). Studies with islets prepared from α-cell-specific A2AR-KO mice confirmed that UK 432097-induced glucagon secretion was mediated by α-cell A2ARs (Fig. 4b).

Somewhat surprisingly, treatment of WT mouse islets with a selective A2AR antagonist (SCH 442416) led to a pronounced reduction in glucagon secretion when glucose levels were low (Fig. 3a). As shown previously, intraislet adenosine[41] and α-cell cAMP[42] levels are high in a low glucose environment. For this reason, the decrease in glucagon levels observed in the presence of SCH 442416 is most likely due to inhibition of glucagon release stimulated by adenosine-mediated activation of α-cell A2ARs. This mechanism also provides a likely explanation for the ability of SCH 442416 to greatly reduce cAMP levels at G3 (Fig. 3c). The glucagon release data shown in Fig. 3b were carried out at 12 mM glucose when intraislet adenosine levels are low[41]. As a result, SCH 442416 had no significant effect on basal glucagon secretion under these conditions.

Under physiological conditions, extracellular adenosine primarily originates from intracellular adenosine, which crosses the cell membrane via specific nucleoside transporters[39]. As a result, changes in cytoplasmic adenosine concentrations lead to altered extracellular adenosine levels. In theory, extracellular adenosine could also be generated from ATP released from β-cells or other cell types present in pancreatic islets. Despite lacking ecto-5′ nucleosidase, mouse β-cells may also serve as a potential source of intraislet adenosine generated by the breakdown of intracellular ATP[8].

Also, at low glucose levels which favor glucagon release, inactivation of adenosine by the enzyme ADA led to impaired glucagon secretion from control islets (Fig. 3e) but not from islets lacking α-cell $G\alpha_s$ or from islets derived from α-A2AR-KO mice (Fig. 3e). These data clearly indicate that islet adenosine acts as an autocrine or paracrine factor that promotes glucagon release by activating the α-cell A2AR/$G_s$ signaling cascade when glucose levels are low.

In this context, it is important to emphasize that extracellular adenosine concentrations in pancreatic islets are high when glucose levels are low but low when glucose levels are high[41]. Yang et al.[41] reported that extracellular adenosine levels at 3 mM glucose are ~6 μM in mouse islets but much lower at high glucose concentrations (10 mM or higher). Since adenosine exhibits nanomolar affinity for A2ARs[62], 6 μM adenosine is predicted to efficiently stimulate endogenous α-cell A2ARs.

Reduced blood glucose levels are known to lead to enhanced cytoplasmic ADP levels (increase in ADP/ATP ratio), the primary stimulus for the activation of AMP-selective 5′-nucleotidase, the enzyme

that promotes the formation of intracellular adenosine[63]. These findings provide a plausible explanation for the inverse relationship between extracellular adenosine levels detected in pancreatic islets and the surrounding glucose concentration. In sum, the present study, together with published data[41], strongly suggests that low blood glucose levels result in increased intraislet adenosine levels which act on α-cell A2ARs to enhance glucagon secretion.

Receptor-mediated activation of $G_s$ results in the activation of adenylyl cyclase, leading to elevated intracellular cAMP levels[64]. As reviewed recently[52,65], elevated intracellular cAMP levels play a key role in promoting glucagon secretion from pancreatic α-cells. Various lines of evidence suggest that this cAMP response requires the activity of protein kinase A (PKA) and Epac2, a guanine nucleotide exchange factor for Rap GTPases[52,65].

To explore the metabolic consequences of inactivating the α-subunit of $G_s$ in α-cells of adult mice, we generated and analyzed α-GsKO mice. As expected, studies with isolated islets showed that the lack of α-cell $G\alpha_s$ virtually abolished the ability of GPCR agonists acting on α-cell β-adrenergic, $A_{2A}$ adenosine, and GIP receptors (isoproterenol, UK 432097, and GIP, respectively) to promote glucagon release (Figs. 5d, e; 4b, 7h). In contrast, treatment of α-GsKO islets with a V1bR agonist (d[Leu⁴,Lys⁸]VP) still resulted in a significant stimulation of glucagon secretion (Fig. 5f; G12). In contrast to the β-adrenergic, $A_{2A}$, and GIP receptors, the V1bR is a $G_{q/11}$-coupled receptor that releases glucagon by activating α-cell $G_{q/11}$ signaling[35]. The ability of the V1bR to promote significant glucagon secretion in the absence of α-cell $G\alpha_s$ clearly indicates that there is still a substantial amount of releasable glucagon left in α-GsKO mice (Fig. 5h). Moreover, KCl retained the ability to stimulate glucagon section in α-GsKO islets, although with reduced efficacy (see, for example, Figs. 5f, 7h). Moreover, pancreata from α-GsKO mice still contain a considerable amount of glucagon (Fig. 5h) which can serve as the source of glucagon in response to non-$G_s$-dependent stimuli. These observations clearly indicate that the relative inability of GPCR agonists acting on α-cell β-adrenergic, $A_{2A}$, and GIP receptors to promote glucagon release from α-Gs KO islets is due to the selective inactivation of α-cell $G_s$ signaling, in combination with the reduced amount of glucagon stored by α-Gs KO islets.

Interestingly, a recent study[36] showed that treatment of mice with a GIP/alanine mixture mimics the major metabolic effects of a mixed meal. In agreement with studies carried out with isolated islets (Fig. 7h), i.p. treatment of control mice with a GIP/alanine mixture resulted in an increase in both plasma glucagon and insulin levels (Fig. 7e, f). Strikingly, this effect was abolished in α-GsKO mice (Fig. 7e, f), suggesting that α-cell $G_s$ signaling is required for the release of glucagon and insulin after consumption of a mixed meal.

Although α-GsKO mice showed a significant decrease in plasma glucagon levels (Fig. 6a, d), this deficit did not cause any significant changes in plasma insulin levels (Fig. 6b, e). We speculate that chronic hypoglucagonemia causes compensatory changes involving other factors and neuronal pathways that can maintain normal plasma insulin levels. In agreement with this notion, previous studies demonstrated that the near-total ablation of α-cells or suppression of α-cell glucagon expression does have any discernible effect on plasma insulin levels in vivo[66–68]. In contrast, acute lowering of plasma glucagon levels due to activation of an inhibitory DREADD expressed in mouse α-cells led to impaired insulin release in vivo[13].

We also made the surprising observation that α-GsKO islets showed a pronounced reduction in pancreatic and islet glucagon content (Fig. 5g, h), while pancreatic or islet insulin content remained unaffected by α-cell $G\alpha_s$ deficiency (Fig. 5i, j). Gene expression analysis showed that Gcg mRNA levels were significantly reduced in α-GsKO islets (Fig. 5k), providing an explanation for the finding that pancreatic glucagon content was reduced in the absence of α-cell $G_s$ signaling. It is highly likely that the observed reduction in pancreatic glucagon content makes a major contribution to the functional impairments

displayed by the α-GsKO islets. However, UK 432097- and isoproterenol-stimulated glucagon secretion was absent or nearly abolished in α-GsKO islets (Figs. 3f, g and 5d, e, respectively). These data indicate that the lack of α-cell $G_s$ signaling also contributes to the functional deficits caused by α-cell $G_s$ deficiency.

The activity of the rodent *Gcg* promoter is under the control of several regulatory factors, including a cAMP response element (CRE) reviewed in refs. 69,70. The promoter of the rodent *Gcg* gene contains a CRE sequence that is activated by cAMP-dependent protein kinase A, resulting in enhanced *Gcg* transcription[58,59]. This observation is consistent with our finding that impaired α-cell $G_s$/cAMP signaling leads to reduced *Gcg* RNA levels (Fig. 5k) and decreased pancreatic glucagon content in α-GsKO islets (Fig. 5g, h). The latter finding also provides an explanation for the observation that treatment of α-GsKO islets with KCl or with an agonist that acts on $G_q$-coupled V1bRs resulted in impaired increases in glucagon secretion (Figs. 5f, 7h). In agreement with these published data[58,59], we showed that treatment of mouse α-TC6 cells with PKI 14-22, a highly selective inhibitor of PKA, resulted in significantly decreased *Gcg* RNA levels (Supplementary Fig. 7g). In sum, these findings highlight the importance of basal α-cell $G_s$ signaling in maintaining proper *Gcg* transcription and glucagon content in pancreatic islets.

Glucagon is the main hormone that helps restore euglycemia in response to hypoglycemic or glucopenic conditions[2,9,10]. We found that the increases in plasma glucagon levels caused by insulin-mediated hypoglycemia or 2-DG-induced glucopenia were significantly reduced in α-GsKO mice, as compared to control littermates (Fig. 7a, c). This observation supports the concept that α-cell $G_s$ signaling plays an important role in mediating the counter-regulatory stimulation of glucagon secretion during hypoglycemic or glucopenic states. However, since α-GsKO mice showed a reduction in pancreatic glucagon content (see previous paragraph), this latter deficit may contribute to the impaired counter-regulatory increases in plasma glucagon levels displayed by α-GsKO mice after insulin or 2-DG treatment.

In conclusion, by analyzing several newly generated mutant mouse models, we demonstrated that α-cell $G_s$ signaling plays a major role in promoting glucagon release and *Gcg* transcription in vivo. We also made the interesting observation that intraislet adenosine can act as an autocrine or paracrine factor to stimulate α-cell A2ARs that are expressed by α-cells at relatively high levels. Glucagon is known to regulate numerous important physiological processes[2,5]. Moreover, since impaired regulation of glucagon release plays a key role in the pathophysiology of diabetes[6–8], the outcome of this study may accelerate the development of strategies able to modulate the activity of pancreatic α-cells for therapeutic purposes.

## Limitations of the study
One limitation of the current study is that α-GsD mice only received acute injections of the DREADD agonist, DCZ. We are planning to explore the metabolic outcome of chronic DCZ treatment of lean and obese (diabetic) α-GsD mice in future studies. It should also be noted that glucagon does not only modulate glucose homeostasis but also has many other important physiological functions, including the suppression of appetite, enhanced energy expenditure, and increased protein catabolism[2,5]. Exploring the potential role of enhanced or reduced α-cell $G_s$ signaling in modulating these additional glucagon functions will be the subject of future work.

## Methods
### Study approval
All animal studies were approved by the National Institute of Diabetes and Digestive and Kidney Diseases Institutional Animal Care and Use Committee. The University of Pennsylvania Institutional Review Board exempted research with human islets from ethical review because the islets were received from deceased, deidentified organ donors. All pancreata were from deceased donors with consent from their families through the United Network for Organ Sharing.

### Drugs, reagents, commercial kits, and antibodies
All drugs, reagents, commercial kits, and antibodies and their sources are listed in Supplementary Table 1.

### Mouse maintenance and diet
All mice were housed in a controlled environment (23 °C, 12-h light/12-h dark cycle) with ad libitum access to food. The animals were maintained on either standard (regular) mouse chow (7022 NIH-07, 15% kcal fat, energy density 3.1 kcal/g, Envigo Inc.) or a high-fat diet (HFD; F3282, 60% kcal fat, energy density 5.5 kcal/g, Bioserv). Male WT mice aged 8–12 weeks were obtained from Taconic (C57BL/6NTac mice). In vivo studies were performed with male mice that were at least 10 weeks old, unless stated otherwise. For in vitro studies, experiments were performed using islets isolated from 12 to 24-week-old male and female mice. Mice consuming HFD for at least 8 weeks were subjected to metabolic tests starting at the age of 14–27 weeks.

All animals were kept under conditions that minimized stress (e.g., proper housing conditions, enrichment strategies, etc.), according to the Guidelines of the NIH Animal Research Advisory Committee (ARAC). Moreover, to reduce injection-induced stress, mice were handled daily for one week including i.p. vehicle injections prior to performing actual experiments involving i.p. injections.

### Generation of α-GsD mice, α-GsKO, and α-A2AR-KO mice
To generate α-cell-specific Gs-DREADD mice (α-GsD mice), heterozygous *ROSA26-LSL-Gs-DREADD-CRE-luc* mice (short name: *LSL-GsD* mice)[33] were crossed with heterozygous *Gcg-Cre^ERT2* mice[34], resulting in *LSL-GsD Gcg-Cre^ERT2* mice. To induce GsD expression in α-cells of these mice, *LSL-GsD Gcg-Cre^ERT2* mice (age: 6–8 weeks) were injected i.p. with TMX for 5 consecutive days (2 mg per mouse per day dissolved in corn oil). TMX-injected Cre-negative *LSL-GsD* mice were used as control mice. All experiments were conducted at least 4 weeks after the last TMX injection to ensure that gastrointestinal L-cell lacked GsD expression.

To obtain α-cell-specific A2AR-KO mice, we crossed *Adora2a^fl/fl* mice[40] with *Gcg-Cre^ERT2* mice[34], yielding *Adora2a^fl/fl Gcg-Cre^ERT2* mice. These mice were subjected to the same TMX treatment protocol as outlined in the previous paragraph. TMX-treated *Adora2a^fl/fl Gcg-Cre^ERT2* mice are referred to as A2AR-KO mice throughout this study. TMX-injected Cre-negative *Gnas^fl/fl* mice lacking the *Gcg-Cre^ERT2* transgene served as control animals.

To generate α-cell-specific α-GsKO mice, we crossed *Gnas^fl/fl* mice[51] with *Gcg-Cre^ERT2* mice[34]. The resulting *Gnas^fl/fl Gcg-Cre^ERT2* were treated with TMX as indicated above. We refer to the TMX-treated *Gnas^fl/fl Gcg-Cre^ERT2* mice as α-A2AR-KO mice throughout this study. In all experiments, TMX-injected Cre-negative *Gnas^fl/fl* mice were used as control mice.

All mice used in this study were maintained on a C57BL/6 background.

### Mouse genotyping
Mouse tail DNA was used for detecting the presence the *Gcg-Cre^ERT2*, *ROSA26-LSL-Gs-DREADD-CRE-luc*, *Gnas^fl/fl*, and *Adora2a^fl/fl* alleles. PCR primers used for genotyping studies are listed in Supplementary Table 2. PCR reactions were carried out using standard procedures[71].

### Hormone and metabolite measurements
Blood samples for hormone and metabolite measurements were collected from the tail vein in EDTA-coated tubes (SAFE-T-FILL, RAM Scientific) containing aprotinin (500 KIU/mL), dipeptidyl peptidase-4 inhibitor (KR-62436, 0.01 mM) and proteinase inhibitors cocktails

(cOmplete Protease Inhibitor Cocktail, Millipore Sigma). Plasma was obtained by centrifugation at 10,000 $g$ for 10 min at 4 °C and stored at −80 °C until analysis. Commercial ELISA kits were used to measure plasma insulin, glucagon, SST, GLP-1, and GIP levels. All kits used for hormone and metabolite measurements are listed in Supplementary Table 1. To measure pancreatic glucagon, insulin, and SST content, we first measured pancreas weight. Pancreata were homogenized in acid-ethanol (1.5% HCl in 70% EtOH, 3 ml/pancreas) using a Precellys Evolution Touch homogenizer (Bertin Instruments) and then incubated on ice for 1 h. To measure the glucagon and insulin content of pancreatic islets, 10–20 hand-picked islets were subjected to sonication in acid-ethanol solution (500 µl/sample). Tissue or islet homogenates were incubated overnight at 4 °C and then centrifuged at 10,000 $g$ for 20 min at 4 °C. Glucagon and insulin levels were measured in the supernatant.

### In vivo metabolic studies
All metabolic tests were performed with adult mice that were at least 10 weeks old (age range: 10–30 weeks) using standard protocols. Unless stated otherwise, male littermates were used for in vivo studies. Blood glucose and plasma insulin and glucagon levels were measured in both freely fed mice at 9 a.m. or in mice that had been fasted overnight for 14 h (time of measurement: 9 a.m.). To assess the in vivo effects of acute activation of the GsD signaling in α-cells, freely fed α-GsD mice and their control littermates were injected with either DCZ (10 µg/kg in saline, i.p.) or saline alone. For i.p. glucose tolerance tests (ipGTT), mice that had been fasted for 14 h were injected with either 2 g glucose/kg (mice consuming regular chow) or 1 g glucose/kg (mice maintained on HFD), respectively. For insulin tolerance tests (ITT), mice that had been fasted for 4 h were injected i.p. with either 0.75 IU/kg (mice on regular chow) or 1 IU/kg/kg (mice on HFD), respectively. To induce glucopenia, mice were injected with 2-DG (500 mg/kg, i.p.) after a 4–5 h fast. To measure glucose-stimulated insulin secretion (GSIS), mice were fasted overnight for 14 h and then injected i.p. with either 1 or 2 g/kg of glucose, as indicated throughout the text. Blood glucose levels were measured prior to injections and at defined post-injection time points using the Contour Blue glucometer (Bayer). Blood was collected from the tail vein. To study the effect of GIP and alanine on glucagon secretion in vivo, mice were fasted for 5 h and then injected i.p. with [D-Ala²] GIP (4 nmol/kg) and/or alanine (0.325 g/kg). Blood was collected in EDTA-coated tubes at specific time points, and plasma hormones were detected using specific ELISA kits (Supplementary Table 1), according to the manufacturers' instructions.

To study the effect of UK 432097 or PSB 0777 (A2AR agonist) on plasma glucagon levels, α-A2A-KO or α-GsKO mice and their corresponding control littermates were fasted for 4 h and then injected i.p. with UK 432097 (5 mg/kg) or PSB 0777 (1 mg/kg). Blood samples were collected at specific post-injections time points, followed by the measurement of blood glucose and plasma hormone levels (see above).

### Isolation of pancreatic islets
Pancreatic islets were prepared from male or female mice (mouse age: 14–24 weeks)[72]. Islets were collected in culture medium (RPMI 1640 medium with 1% penicillin/streptomycin, 10% fetal bovine serum [FBS], and 11 mM glucose). The isolated islets were used for islet perifusion, flow cytometry, RNA isolation, and the determination of glucagon and insulin content.

### Islet perifusion studies
Islet perifusion studies were performed with handpicked islets[73]. After incubating islets overnight in culture medium (RPMI1640, 10% FBS, and 1% penicillin/streptomycin), islets were placed into perifusion chambers (75–100 islets/chamber) with Bio-Gel P-4 media (Bio-Rad) to immobilize islets in an automated perifusion system (Biorep Perifusion

System). All compounds used were dissolved in perifusion buffer (composition in mM: 125 NaCl, 5.9 KCl, 2.56 CaCl$_2$, 1.2 MgCl$_2$, 25 HEPES, and 0.1% BSA, pH 7.4). A peristaltic pump pushed reagents continuously into the islet-containing chambers. Perfusates were collected in ice-cold 96-well plates for further analysis after equilibration for 48 min in either 3 or 12 mM glucose. Glucagon and insulin concentrations in the perfusates were determined with specific ELISA kits (see Supplementary Table 2).

### Static glucagon secretion assay
Isolated islets were incubated in culture medium for 2 h. Batches of 10 islets from control and α-GsKO mice were pre-incubated in 0.5 ml of "static buffer" (KRB containing 1 mg/ml BSA and 3 or 12 mM glucose, respectively) for 1 h in a cell culture incubator at 37 °C. Subsequently, islets were transferred to 0.5 ml static buffer for 1 h and then incubated in 0.5 ml static buffer for 1 h in the absence or presence of drugs. After each incubation, the supernatant was removed to measure glucagon secretion, and 0.25 ml of acidified ethanol (vol/vol: 75% ethanol/1.5% HCl) was added to the islets to determine total islet glucagon. Glucagon concentrations were measured by using a glucagon ELISA kit (Lumit Glucagon Immunoassay).

### Flow cytometry
Following islet isolation, islets were allowed to recover overnight. 70–100 islets were hand-picked from each pancreas and rinsed in PBS before incubation in Accutase (Sigma, A6964) at 37 °C for 12–15 min with occasional vortexing. Subsequently, cold RPMI medium was added to terminate the digestion process, and cells were centrifuged for 3 min at 350 $g$ at 4 °C. The medium was then aspirated, and islet cells were washed with sorting buffer [RPMI 1640 without phenol red (11835030), 11.1 mM glucose, 1% FBS, 1% penicillin/streptomycin, 20 mM HEPES, and deoxyribonuclease (10 U/ml)]. After this step, cells were filtered through a 30-µm mesh and sorted using a Beckman-Coulter MoFlo Astrios Cell Sorter using forward and side scatter to separate single cells from debris and doublets. Finally, cells were separated with side scatter and autofluorescence into enriched populations of α- β-, and δ-cells into Trizol.

### α-cell cAMP and calcium imaging
α-CAMPER mice were generated by crossing CAMPER mice (Jax 032205), a Cre-dependent cAMP indicator strain[74], with *Glucagon-Cre^ERT2* mice (Jax 030346) that express tamoxifen-inducible Cre^ERT2 in islet α-cells[34]. α-GCaMP6s mice were obtained by crossing GCaMP6s mice (Jax 028866), a Cre-dependent Ca²⁺ indicator strain[75], with *Glucagon-Cre^ERT2* mice (Jax 030346)[34]. Imaging of α-cell cAMP and Ca²⁺ was performed using epifluorescence microscopy of intact islets as detailed in ref. [36], with glucose and drug applications indicated in the figure legends.

### Western blotting
For western blotting studies, mouse tissues (~20 mg) were homogenized in 400 µl of ice-cold RIPA buffer containing a protease inhibitor cocktail (cOmplete Protease Inhibitor Cocktail, Millipore Sigma). Protein concentrations were determined by a BCA assay. Protein extracts (20 µg) were incubated at 37 °C for 10 min in NuPAGE LDS sample buffer, separated using 3–8% Tris-acetate SDS-PAGE gels (Thermo Fisher Scientific), and then blotted onto nitrocellulose membranes (Bio-Rad). Membranes were blocked for 1 h at room temperature in blocking buffer (5% BSA, 0.1% Tween 20 in PBS) and incubated overnight at 4 °C with primary antibody in blocking buffer. HRP-conjugated anti-rabbit or anti-mouse secondary antibodies and SuperSignal West Pico Chemiluminescent Substrate (Pierce) were used to visualize protein bands via an Azure 300 Imaging System Imager (Azure Biosystems). Detailed information regarding the antibodies used is given in Supplementary Table 1.

## Preparation of brain slices

Control and α-GsD mice were perfused through transcardiac perfusion with 20 ml of saline followed by 50 ml of 4% paraformaldehyde (PFA). Mouse brains were collected and incubated in 4% PFA overnight. Subsequently, brains were transferred to 30% sucrose solution. Brains were sectioned (section thickness: 30 μM) with a Vibratome or a Cryostat (for brain stem immunofluorescence studies). We prepared sections covering the whole nucleus tractus solitarius (NTS, AP −6.50−8.00 mm).

## Immunofluorescence studies

To detect the expression of the GsD designer receptor or Gα$_s$ in mouse pancreatic islets, pancreata were fixed overnight with 4% paraformaldehyde, embedded in paraffin, and then sectioned (section thickness: 5 μm). After deparaffinization and rehydration, sections were subjected to heat-induced antigen retrieval (low pH, eBioscience) and then treated with blocking buffer (5% goat serum, 0.1% Tween 20 in PBS) for 1 h at room temperature. To detect the expression of the GsD in the brain, brain sections were incubated in blocking buffer for 1 h at room temperature. Subsequently, all sections were stained with primary antibody (overnight incubation) and secondary antibodies carrying conjugated fluorophores. Fluorescence images were acquired using a confocal microscope (Zeiss LSM 700). Details regarding the antibodies used are provided in Supplementary Table 1.

## Islet morphometric studies

Pancreatic sections were prepared as outlined in the previous paragraph. To measure α- and β-cell mass, the areas of the regions staining positive for insulin or glucagon and the total area of each section were quantified with a Keyence digital microscope (BZ-9000) with a CFI Plan Apo λ ×4 lens. The ratio of the hormone-stained area to the area of the total pancreatic section was averaged for each mouse and then multiplied by pancreas weight. For these studies, 3 or 4 mice (females) per genotype were used.

## Culture of α-TC6 cells

α-TC6 cells (kindly provided by Dr. Rohit Kulkarni at Harvard Medical School) were cultured in DMEM supplemented with 10% FBS, 10 mg/ml penicillin, and streptomycin at 37 ℃ and 5% $CO_2$. Cells were treated with a selective PKA inhibitor (PKI 14-22, 10 μM) for 16 h and then harvested with 1 ml Trizol. Subsequently, RNA was extracted and subjected to qRT-PCR studies.

## Analysis of mRNA expression levels via qRT-PCR

To measure mRNA expression levels, RNA was extracted from TRIzol lysates prepared from either whole mouse pancreatic islets or from FACS-sorted islet cells using an RNA miniprep kit according to the manufacturer's instructions (Direct-zol RNA Miniprep kit, R2053, Zymo Research). cDNA was synthesized using SuperScript III First-Strand Synthesis SuperMix (Invitrogen). Quantitative PCR (qPCR) studies were performed under standard conditions using gene-specific primers (for details, see Supplementary Table 2) and SYBR Green (Luna Universal qPCR Master Mix, M3003, NEB) or TaqMan reagents. Data were analyzed by calculating ΔΔCt values. The expression of each gene of interest was normalized to the expression of cyclophilin A or 36b4 (encoded protein: acidic ribosomal phosphoprotein P0).

## Glucagon release studies with perifused human islets

Human pancreatic islets were provided by the NIDDK-funded Integrated Islet Distribution Program (IIDP) (RRID:SCR_014387) at the City of Hope (NIH Grant #U24DK098085). Donor information is provided in Supplementary Table 3. Upon receipt, islets were cultured in Prodo Islet Media (Standard) (Prodo Laboratories) for three days before perifusion using the PERI-4 system (Biorep Technologies). One hundred islets were placed in each chamber and perifused for 30 min in substrate-free Krebs buffer (114 mM NaCl, 5 mM KCl, 24 mM NaHCO$_3$, 1 mM MgCl$_2$, 2.2 mM Ca$^{2+}$, 1 mM P$_i$, 10 mM HEPES, 0.25% BSA, pH 7.4). Glucose (3 or 12 mM) was then added for 20 min to allow glucagon secretion to equilibrate before adding the A2AR antagonist SCH 442416 (1 μM) and/or the A2AR agonist UK 432097 (5 nM). Flow rate was set at 100 μl/min, and samples were collected every minute. Glucagon secretion was measured via ELISA (see Supplementary Table 1).

## Statistics

All data are expressed as means ± SEM for the indicated number of observations. Statistical significance was evaluated by employing GraphPad Prism 9 software (La Jolla, CA). A $P < 0.05$ was considered statistically significant. The statistical tests and sample sizes used in the individual experiments are described in the figure legends.

## Reporting summary

Further information on research design is available in the Nature Portfolio Reporting Summary linked to this article.

## Data availability

All data generated in this study are provided in the Supplementary Information/Source Data file. Source data are provided with this paper.

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

## Acknowledgements
The *Adora2a^{fl/fl}* mice[40] were kindly provided by J. Linden and Y. Huo. The *CAG-LSL-GsD* mice were generated and made available to us by R. Berdeaux[33]. This research was funded by the Intramural Research Program of the National Institute of Diabetes and Digestive and Kidney Diseases (NIDDK, NIH). K.E. was supported by funding from the NIH/NIDDK (K01 DK132461). S.M.G. was supported by an NIH training grant (F32DK121420). J.E.C. gratefully acknowledges support from the NIH/NIDDK (R01 DK123075, DK125353, DK046492) and the Helmsley Charitable Trust Foundation. J.R. and N.D. thank the University of Pennsylvania Diabetes Research Center for access to the Islet Cell Biology Core and Radioimmunoassay Core (P30-DK19525). The Merrins laboratory gratefully acknowledges support from the NIH/NIDDK (R01DK113103 and R01DK127637 to M.J.M., and F31DK134171 to E.R.K.), and the United States Department of Veterans Affairs Biomedical Laboratory Research and Development Service (I01B005113 to M.J.M.). We thank Zhenzhong Cui (NIH/NIDDK) for his help with the immunostaining studies and Sungyoung Auh (NIDDK Biostatistics Program) for advice regarding the proper statistical analysis of our experimental data.

## Author contributions
L.L., K.E., J.E.C. and J.W. designed the study. L.L. and J.W. wrote the paper. L.L., K.E., D.D., L.B., Y.C., S.G., C.G., E.K., E.J., M.J.M., J.R., N.D. and J.E.C. carried out experiments and interpreted and analyzed experimental data. M.C., L.S.W. and K.H.K. provided essential tools/resources.

## Funding

## Competing interests
J.E.C. receives funding for basic research from Eli Lilly, Novo Nordisk, and Merck. The other authors declare no competing interests.
