## [Peer Review File · Nature Communications]

Intra-islet α -cell Gs signaling promotes glucagon releaseREVIEWER COMMENTS

Reviewer #1 (Remarks to the Author):

In this study, Liu et al have generated a number of new mouse strains to clarify the role of the Gs protein and Gs-coupled receptors in the regulation of glucose homeostasis and glucagon secretion. Chemigenetic Gs-coupled DREADD receptors were specifically expressed in alpha cells and their activation improved glucose tolerance and promoted both glucagon and insulin secretion in vivo and enhanced hormone release from islets ex vivo in response to alanine and depolarization. Pharmacological and genetic tools were used to demonstrate a role for adenosine A2A receptors to sustain glucagon secretion under hypoglycemic conditions, an effect proposed to depend on auto- or paracrine effects from adenosine released within islets. Consistent with the importance of Gs-coupled receptor signaling, lack of the Gs protein in alpha cells was associated with impaired glucagon secretion as well as reduced glucagon gene transcription.

This is a sound and interesting study providing valuable new insights into the regulation of glucagon secretion. An extensive amount of data is presented to support the conclusions. The methodology is solid, the experiments are overall well-designed and the results clearly described. There are nevertheless some points that should be considered for improving the manuscript.

1. P 5: The reader may get the impression that the authors discovered the selective expression of A2AR in alpha cells, but this information was actually picked up from already published data. Moreover, as can be seen in Fig 4a and Suppl Fig 3a, the Adora2 receptor is expressed also in pancreatic delta cells (to almost the same extent as in alpha cells), which may be worth mentioning.
2. The immunofluorescence images in Suppl Fig 1b is too dim, at least in the pdf version of the manuscript. That quality is insufficient and must be improved.
3. P 8: The authors claim that DCZ enhanced glucagon secretion at both 3 and 12 mM glucose but this is not evident from Fig 1g. The difference exists already before application of DCZ and the effect of acute application of the drug is not at all evident and no statistics is presented to support the statement.
4. P 8, 3rd line from the bottom: It is mentioned that GsD shows enhanced basal activity in vitro, but Fig 1d indicates that there is also higher basal glucagon levels in vivo.
5. P 11, 2nd para: Treatment of A2AR agonist is said to have little effect on cAMP at low glucose but from Fig 3c it seems that there is reduction, similar in magnitude to the increase observed at 12 mM glucose.
6. P 11, 2nd para: The authors speculate that cAMP levels in alpha cells are already high at low compared to high glucose. Studies of the effects of glucose on alpha cell cAMP has shown that it is indeed the case, and the authors may consider to include a reference, such as PMID 30953108.
7. In Figs 3c and 4b and 4c, no or only one data point is shown before the application of the test stimulus, which makes it impossible to evaluate the effects.
8. P 15/Fig 4g,h: 2-DG administration triggers glucagon secretion, which is impaired in the α -A2AR-KO. Yet, the resulting hyperglycemia is more pronounced in the KO, despite similar insulin levels. This seemingly confusing observation deserves to be discussed.
9. Investigating the role of Gs in the α -GsKO, the authors focused on beta adrenergic signaling using isoproterenol. That is interesting but it would have been logical to also test an A2AR agonist to consolidate the findings from the A2AR-KO islets.
10. The authors show that Gs signaling is crucial for GIP effects on glucagon secretion. Did

the authors also investigate GLP-1? The mechanism underlying the inhibition of glucagon secretion by GLP-1 is controversial and the tools generated by the authors would provide valuable mechanistic insights.

11. The discussion about the source of adenosine in the islet is rather vague and could be improved. It seems as the authors favor an autocrine mechanism (“In theory...adenosine could be generated from beta cells”; “However, ... the enzyme that catalyzes ...the conversion of ATP into adenosine is absent in mouse and human islets”). Even if mouse and human islets lack the ectonucleotidase that converts ATP to adenosine, beta cells may be the source.

Minor:

12. Page 4, second para: The phrasing “direct glucose sensing is a key determinant of glucagon secretion” is somewhat unclear. Glucose “sensing” takes place both by neural mechanisms and by multiple cell types in the islets. Why not just “glucose is a key determinant”?

13. P 4, third para: “dozen” should probably read “dozens”.

14. P 8: “Incubation” does not seem to be a correct description. “Exposure” or “perifusion” would be more appropriate for the protocol shown in Fig 1g.

15. P 22, bottom line: Only Fig 1g and not 1h shows glucagon secretion.

16. On p 24, 1st paragraph: It is questionable if a study from 2010 should be called recent.

Reviewer #2 (Remarks to the Author):

The authors of this manuscript explored a role for Gs in alpha-cells of pancreatic islets. They used a Gs-DREADD selectively expressed in these cells in a transgenic mouse model to probe the increasingly complicate physiology of the glucagon system. They also use an alpha-cell specific Gs knockout line of mice. The manuscript is well written in general and was a pleasure to read.

The characterization of the mouse models was appropriate and leads to confidence in asserting that they were alpha-cell specific. Controls for this were provided in the supplemental material. The experiments are well conducted and the conclusions well supported by the data provided.

Comments

1) In the constitutive DREADD activity a potential confound? Might there be some homeostatic mechanisms that come into play because of this?

2) In Figure 3a, why did SCH 442416 cause a decrease in glucagon levels on its own? Is there any indication that it is acting as an inverse agonist or a biased agonist for other pathways? This effect was also noted when measuring cAMP levels. Why was this effect not seen in Figure 3B?

3) Why weren't Figure 3c and 3d conducted in an identical way?

4) The immunofluorescence data in Figure 5c is not convincing. How about western blots?

5) Why didn't the authors stimulate cells in Figure 5d and 5e with the adenosine receptor

agonist?

6) In figure 6, why aren't there homeostatic alterations in insulin levels when glucagon levels are reduced?

Reviewer #3 (Remarks to the Author):

The manuscript entitled “Novel mouse models establish a key metabolic role for α -cell Gs signaling” by Liu et. al demonstrate that Gs-coupled signaling impacts α -cell function. Understanding alpha cell function and the regulation of glucagon secretion could have important implications for diabetes treatment so the topic is timely and relevant. The authors use chemogenetics to activate G α s signaling within Gcg cells in lean and obese animals and study in vivo and ex vivo glucagon and insulin secretion. DREADD activation in both lean and obese animals stimulated glucagon and insulin secretion and improved IP glucose tolerance. The authors also employ several mouse models that disrupt Gs signaling (α -G α sD and G α s) or Gs adenosine signaling (A2AR) and show that the A2A adenosine receptors play a role in stimulating glucagon secretion under low glucose conditions. Although there are interpretive issues, the authors state that the perfusion studies agree. Long-term disruption of G α s signaling lowers Gcg expression, islet glucagon content, and consequently basal and glucagon responses to various stimuli. The authors conclude that G α s is a novel target for regulating glucagon secretion. Although there is enthusiasm for the topic of research, there are weaknesses associated with the mouse model, the statistical analysis, and some issues with data interpretation.

Major comments:

1. The Gcg Cre utilized here has not been validated for brain expression. Gcg neurons are known to be expressed in the hindbrain and function to regulate various physiological functions including feeding patterns, body weight in some cases, motivation for feeding etc. More and more studies are being published on the impact of these neurons on physiology. Therefore, validation that the genetic manipulation has not targeted the brain is required. They easily could have done the HA tag staining should have been completed in the nucleus of the solitary tract-why it was done in the hypothalamus instead is unclear.
2. Were the data in Figure 1 analyzed with an ANOVA for repeated measures? If there is a group effect, but no time effect then statistically you can't say that DCZ actually increased glucagon or insulin. The statistics need to be redone with an ANOVA for repeated measures with an appropriate post hoc analysis for all graphs where time is a variable.
3. Figure 1: Although it is not clear whether this is statistically significant, there is a 50% drop in glucagon with saline injection and with DCZ in the control animals. This drop is prevented in the G α sD animals with DCZ-again it is not clear whether DCZ actually significantly increases glucagon at 15 min or whether it is simply preventing a drop in glucagon. This finding is very curious and needs further exploration/explanation.
4. Figure 1: Is there a statistical increase in glucose with the saline or DCZ injection? If so, then these studies are likely confounded by the animal's stress levels.
5. Figure 1 and Supp Figure 1, perfusion studies: Is the relative increase in glucagon with the Ala or AAM greater than the control animals? It seems not. Thus, the increase in glucagon is simply due to an increase in basal concentrations. If this is the case then there seems to be an ex vivo and in vivo difference in glucagon responses that needs to be discussed. Further, it is unclear why a doubling of glucagon under “basal conditions” is visible in perfused islets but not in vivo. The fact that the insulin is normal at baseline and is higher with amino acid stimulus even though the relative increase in glucagon is similar

dissociates glucagon and insulin responses to amino acids under these conditions. This becomes hard to see with the expanded scale in Figure 1g. The components of the perfusion experiments should be separated out so the data can be visualized on an appropriate scale for Fig 1g. In fact, is it the scale or did the control alpha cells not increase glucagon secretion in response to alanine?

6. Statistical analysis of the perfusion studies needs to be reported.

7. Body weight of Adora2af/f Gcg cre mice is needed.

8. Interpretation of the 2dg data in the alpha-A2AR KO mice is not accurate. These mice had decreased glucagon and increased glucose responses to 2DG. Under glucoprivic conditions, increases in glucagon function to stimulate hepatic glucose production and increase plasma glucose. Since insulin levels are the same, then this clearly indicates that the insulin to glucagon ratio should favor a suppression of hepatic glucose production. Therefore, some other counterregulatory factor is regulating the glucose response to 2DG in the face of inadequate glucagon levels in these animals. This suggests off-target impact on sympathetic neurons. Plasma catecholamines, which also potently respond to 2DG and regulate glucose homeostasis should be assessed.

9. There are more 2DG interpretive issues in Figure 7. Specifically in the second part of this sentence: "The glucopenia-induced increases in blood glucose levels were significantly more pronounced in a-GsKO mice (Fig. 7d), probably due to decreased plasma insulin levels (Supplementary Fig. 7a, b)." The 2dg-induced (and it's not 2DG it's the increase in glucose that drives the increase in insulin) insulin secretion was not reduced, it simply returned to baseline quicker. Thus, the conclusion that α -cell Gs signaling contributes to the counter-regulatory glucagon responses is not accurate. This is another specific example where a repeated measures anova for time is necessary so that the responses over time can be accurately assessed and interpreted.

10. The interpretation of the glucagon responses in the alpha-GsKO mice is also curious. Gcg expression is decreased, islet glucagon content is decreased. So if there is limited gene and protein, then it is not surprising that there is no glucagon response to glucagon secretagogues. It also brings to question why the KO of alpha cell Gs leads to a decrease in Gcg expression in the first place.

11. It would be interested to know how somatostatin regulation of the alpha cell fits into this story.

Minor Comments:

12. Supp Fig 1, Panel B should show individual channels of all fluorophores used.

01-25-2024

Response to the Referees' comments

NCOMMS-23-26486, revised version

TITLE: " Novel mouse models establish a key metabolic role for α -cell G_s signaling"

by Liu Liu *et al.*

General response by the authors

We would like to thank the three reviewers for their constructive comments and suggestions. We appreciate their time and effort and believe that this manuscript has been significantly improved based upon their suggestions.

Please note that we carried out many additional experiments to address the various comments raised by the reviewers. The additional work that we performed has led to the incorporation of several new figure panels (including Suppl. Figures) into the revised version of the manuscript.

Our response to the comments of the three reviewers (highlighted in yellow)

Reviewer #1

In this study, Liu et al have generated a number of new mouse strains to clarify the role of the G_s protein and G_s -coupled receptors in the regulation of glucose homeostasis and glucagon secretion. Chemigenetic G_s -coupled DREADD receptors were specifically expressed in alpha cells and their activation improved glucose tolerance and promoted both glucagon and insulin secretion in vivo and enhanced hormone release from islets ex vivo in response to alanine and depolarization. Pharmacological and genetic tools were used to demonstrate a role for adenosine A2A receptors to sustain glucagon secretion under hypoglycemic conditions, an effect proposed to depend on auto- or paracrine effects from adenosine released within islets. Consistent with the importance of G_s -coupled receptor signaling, lack of the G_s protein in alpha cells was associated with impaired glucagon secretion as well as reduced glucagon gene transcription.

This is a sound and interesting study providing valuable new insights into the regulation of glucagon secretion. An extensive amount of data is presented to support the conclusions. The methodology is solid, the experiments are overall well-designed and the results clearly described. There are nevertheless some points that should be considered for improving the manuscript.

Thank you for your very positive comments about our manuscript.

1. P 5: The reader may get the impression that the authors discovered the selective expression of A2AR in alpha cells, but this information was actually picked up from already published data. Moreover, as can be seen in Fig 4a and Suppl Fig 3a, the Adora2 receptor is expressed also in pancreatic delta cells (to almost the same extent as in alpha cells), which may be worth mentioning.

As requested by the reviewer, we clarify in the revised manuscript that the expression of the A2AR in α -cells was 'picked up' from published data and that this receptor subtype is also expressed in δ -cells. Specifically, we made the following changes:

Results section (page 11, 2nd paragraph): "Analysis of previously published scRNAseq data from human and mouse islets^{1,2} led to the identification of three G_s-coupled receptors that are selectively expressed in α -cells, as compared to other islet cell types."

In the revised manuscript, we now also mention that the A2AR is also expressed in δ -cells (page 11, center): "This receptor subtype is also expressed at low to moderate levels in mouse islet δ -cells (Supplementary Fig. 3a; Fig. 4a)."

2. The immunofluorescence images in Suppl Fig 1b is too dim, at least in the pdf version of the manuscript. That quality is insufficient and must be improved.

As requested by the reviewer, we now provide a significantly improved version of Supplementary Fig. 1b. For improved clarity, we now show the various channels used.

3. P 8: The authors claim that DCZ enhanced glucagon secretion at both 3 and 12 mM glucose but this is not evident from Fig 1g. The difference exists already before application of DCZ and the effect of acute application of the drug is not at all evident and no statistics is presented to support the statement.

We subjected the data shown in Fig. 1g and all other islet perfusion data to a thorough statistical analysis. To correct for differences in basal hormone secretion, we calculated area of the curve (AOC) values by subtracting the areas under or over the baseline³. This type of analysis is the method of choice when baseline levels between two or more experimental groups differ³. We found that DCZ treatment of α -GsD islets resulted in a significant increase in glucagon secretion only in the presence of alanine (Fig. 1g, h). Under physiological conditions, pancreatic islets are exposed to high levels of circulating amino acids which increase the responsiveness of α -cells to various glucagon secretagogues^{4,5}. This observation may explain why DCZ treatment of α -GsD islets had no significant effect on glucagon release in the absence of alanine (new Fig. 1g, h).

In the new version of the manuscript, we added AOC bars next to Fig. 1g (new Fig. 1h) and subjected these data to a statistical analysis. Moreover, we subjected all other islet perfusion data to the same type of analysis (please see the revised versions of Fig. 1g-j; Fig. 3a, b, e, h-j; Fig. 4b-d; Fig. 5d-f; Fig. 7h, and Suppl. Fig. 1 k, l).

We added the following additional text to the Results section (page 9):

"DCZ (10 nM) treatment of α -GsD islets in the presence of 3 or 12 mM glucose resulted in significant increases in glucagon secretion only in the presence of alanine (3 mM) (Fig. 1g, h). Under physiological conditions, pancreatic islets are exposed to high levels of circulating amino acids which increase the responsiveness of α -cells to various glucagon secretagogues^{4,5}, providing a likely explanation for the inability of DCZ to stimulate glucagon release from α -GsD islets in the absence of alanine (Fig. 1g, h)."

4. P 8, 3rd line from the bottom: It is mentioned that GsD shows enhanced basal activity in vitro, but Fig 1d indicates that there is also higher basal glucagon levels in vivo.

Thank you for raising this point. Although there was a trend towards higher basal plasma glucagon level in α -GsD mice (time 0), this effect failed to reach statistical significance ($p=0.085$; $p=0.085$; Student's t-test at time 0). We now mention this fact in the revised version of the manuscript (page 7, bottom).

5. P 11, 2nd para: Treatment of A2AR agonist is said to have little effect on cAMP at low

glucose but from Fig 3c it seems that there is reduction, similar in magnitude to the increase observed at 12 mM glucose.

We fully agree with the reviewer. We should have addressed this issue in more detail in the original manuscript. We now clarify this matter in the revised version of the manuscript (page 12, 2nd paragraph):

"We found that A2AR agonist treatment (UK 432097, 5 or 20 nM) of α -CAMPER islets resulted in a small reduction in cAMP levels at G3 but caused a significant increase in cAMP accumulation at G12 (Fig. 3c, d). The A2AR agonist-induced decrease in cAMP levels at G3 is probably due to the facts that A2ARs are already strongly stimulated by high endogenous adenosine levels⁶ and that α -cell cAMP levels are already high in a low glucose environment⁷. In agreement with these observations, addition of the A2AR antagonist SCH 442416 (0.5 mM) led to a very robust reduction of cAMP levels at G3 (Fig. 3c), raising the possibility that the inhibitory effect of the A2AR agonist at G3 was caused by A2AR desensitization or the activation of other, yet unknown, signaling pathways that interfere with adenosine-induced cAMP production."

The detailed cellular mechanisms underlying this phenomenon remain to be explored experimentally in future studies

6. P 11, 2nd para: The authors speculate that cAMP levels in alpha cells are already high at low compared to high glucose. Studies of the effects of glucose on alpha cell cAMP has shown that it is indeed the case, and the authors may consider to include a reference, such as PMID 30953108.

Thank you for directing us to this highly relevant paper (new ref. 41). We now cite this work in the revised version of the manuscript (page 12, center, and Discussion section, page 24, bottom).

7. In Figs 3c and 4b and 4c, no or only one data point is shown before the application of the test stimulus, which makes it impossible to evaluate the effects.

Very good point. As requested by the reviewer, we measured glucagon release before the application of the test stimulus using samples stored in the freezer. We modified Fig. 3 c and 4b, c, as requested by the reviewer.

8. P 15/Fig 4g,h: 2-DG administration triggers glucagon secretion, which is impaired in the α -A2AR-KO. Yet, the resulting hyperglycemia is more pronounced in the KO, despite similar insulin levels. This seemingly confusing observation deserves to be discussed.

Like the reviewer, we were also puzzled by these findings. For this reason, we decided to repeat this experiment with a new set of mice that we specifically generated to revisit this issue. Using this new batch of mice, we were unable to replicate the data presented in original version of the manuscript (Fig. 4g, h; probably an artifact). The new data show that 2-DG-induced increases in plasma glucagon, plasma insulin, and blood glucose levels were not significantly affected by the lack of α -cell A2ARs. We therefore replaced Fig. 4g, h (original manuscript) with a new figure displaying these new data (Suppl. Fig. 4q-s) in the revised manuscript).

We also changed the text in the Results section accordingly in the revised manuscript (page 16, center): " We found that 2-DG-induced increases in plasma glucagon, plasma insulin, and blood glucose levels were not significantly affected by the lack of α -cell A2ARs (Supplementary Fig. 4q-s)."

9. Investigating the role of G_s in the α -GsKO, the authors focused on beta adrenergic signaling using isoproterenol. That is interesting but it would have been logical to also test an A2AR agonist to consolidate the findings from the A2AR-KO islets.

To address this point, we treated α -GsKO and control islets with a selective A2AR agonist (UK 432097, 50 nM; revised Fig. 3f, g). We found that UK 432097-induced glucagon secretion was absent or nearly completely abolished in α -GsKO islets at both G3 and G12, respectively. In addition, we carried out vivo experiments with PSB 0777 (1 mg/kg, i.p.), another selective A2AR agonist⁸. In agreement with the in vitro data (new Fig. 3f, g), we found that PSB 0777-induced increases in plasma glucagon levels were abolished in α -GsKO mice, as compared to control littermates (new Supplementary Fig. 6c). These data clearly indicate that A2AR-stimulated glucagon secretion requires α -cell G_s signaling. Appropriate changes were made in the revised manuscript (page 19, top).

10. The authors show that G_s signaling is crucial for GIP effects on glucagon secretion. Did the authors also investigate GLP-1? The mechanism underlying the inhibition of glucagon secretion by GLP-1 is controversial and the tools generated by the authors would provide valuable mechanistic insights.

The reviewer raises an interesting point. In fact, we are planning to carry out additional work focusing on the role of G_s in mediating the actions of GLP-1 on different islet cell types including α -cells in a follow-up study.

11. The discussion about the source of adenosine in the islet is rather vague and could be improved. It seems as the authors favor an autocrine mechanism (“In theory...adenosine could be generated from beta cells”; “However, ... the enzyme that catalyzes ...the conversion of ATP into adenosine is absent in mouse and human islets”). Even if mouse and human islets lack the ectonucleotidase that converts ATP to adenosine, beta cells may be the source.

We fully agree with the reviewer. We therefore modified the part of the discussion that deals with this point as follows (page 25, center):
“Despite lacking ecto-5' nucleosidase, mouse β -cells may also serve as a potential source of inraislet adenosine generated by the breakdown of intracellular ATP⁹.”

Minor:

12. Page 4, second para: The phrasing “direct glucose sensing is a key determinant of glucagon secretion” is somewhat unclear. Glucose “sensing” takes place both by neural mechanisms and by multiple cell types in the islets. Why not just “glucose is a key determinant”?

As requested by the reviewer, we now use the term “glucose is a key determinant” in the revised manuscript (page 4).

13. P 4, third para: “dozen” should probably read “dozens”.

This has been corrected.

14. P 8: “Incubation” does not seem to be a correct description. “Exposure” or “perifusion” would be more appropriate for the protocol shown in Fig 1g.

We replaced the term "incubation" with either "exposure" or "perifusion" at all relevant occasions.

15. P 22, bottom line: Only Fig 1g and not 1h shows glucagon secretion.

This has been corrected.

16. On p 24, 1st paragraph: It is questionable if a study from 2010 should be called recent.

Regarding the 2010 paper: we removed the term "recent".

Reviewer #2

The authors of this manuscript explored a role for Gs in alpha-cells of pancreatic islets. They used a Gs-DREADD selectively expressed in these cells in a transgenic mouse model to probe the increasingly complicate physiology of the glucagon system. They also use an alpha-cell specific Gs knockout line of mice. The manuscript is well written in general and was a pleasure to read.

The characterization of the mouse models was appropriate and leads to confidence in asserting that they were alpha-cell specific. Controls for this were provided in the supplemental material. The experiments are well conducted and the conclusions well supported by the data provided.

Thank you for your very positive general comments.

Comments

1) In the constitutive DREADD activity a potential confound? Might there be some homeostatic mechanisms that come into play because of this?

We agree with the reviewer that the constitutive activity of the Gs DREADD (GsD) somewhat complicates the interpretation of the experimental data. To address this issue, we incorporated the following text into the Discussion section (page 24, top): "We noted that basal glucagon release was markedly increased in α -GsD islets, as compared to control islets (Supplementary Fig. 1k). Previous studies have shown that the GsD designer receptor can signal, to a variable degree, in a ligand-independent fashion^{10,11}. Thus, it is likely that the elevated glucagon levels observed with α -GsD islets in the absence of an activating ligand are most likely caused by the constitutive signaling via α -cell GsD. While this effect was easily detectable in α -GsD islets in vitro, basal plasma glucagon levels were similar in α -GsD mice and control littermates in vivo (Fig. 1a, d). This latter observation suggests that constitutive GsD signaling is unlikely to have a major impact on the outcome of the in vivo studies and that other physiological mechanisms, including, for example, circulating hormones and nutrients, paracrine factors, and neuronal pathways, limit constitutive GsD signaling in vivo."

2) In Figure 3a, why did SCH 442416 cause a decrease in glucagon levels on its own? Is there any indication that it is acting as an inverse agonist or a biased agonist for other pathways? This effect was also noted when measuring cAMP levels. Why was this effect not seen in Figure 3B?

The reviewer raises an interesting point. To address this issue, we incorporated the following text into the Discussion section (page 24, bottom, and page 25, top):

"Somewhat surprisingly, treatment of WT mouse islets with a selective A2AR antagonist (SCH 442416) led to a pronounced reduction in glucagon secretion when glucose levels were low (Fig. 3a). As shown previously, inraislelet adenosine⁶ and α -cell cAMP⁷ levels are high in a low glucose environment. For this reason, the decrease in glucagon levels observed in the presence of SCH 442416 is most likely due to inhibition of glucagon release stimulated by adenosine-mediated activation of α -cell A2ARs. This mechanism also provides a likely explanation for the ability of SCH 442416 to greatly reduce cAMP levels at G3 (Fig. 3c). The glucagon release data shown in Fig. 3b were carried out at 12 mM glucose when inraislelet adenosine levels are low⁶. As a result, SCH 442416 had no significant effect on basal glucagon secretion under these conditions.

We are not aware of any published studies examining the ability of SCH 442416 to function as an inverse agonist or a biased agonist for other pathways.

3) Why weren't Figure 3c and 3d conducted in an identical way?

In Fig. 3c (G3), a subset of samples was incubated with SCH 442416 alone, without the UK 432097 agonist. In Fig. 3d (G12), this control was omitted since the experimental design included a SCH 442416 pre-incubation period (internal control). Omitting the "SCH 442416 alone curve" allowed us to study more islets at 12G, where cAMP measurements are noisier due to paracrine influences from β - and δ -cell signaling (Merrins et al, unpublished data).

4) The immunofluorescence data in Figure 5c is not convincing. How about western blots?

To address this point, we modified Fig. 5c by enlarging specific areas to better visualize the presence (or absence) of G_{α_s} protein in α -cells. Western blotting studies are problematic since they would require relatively large amounts of FACS-sorted α -cells which only represent a small subpopulation of all islet cells. However, the qRT-PCR data shown in Fig. 5a convincingly demonstrate that mRNA coding for G_{α_s} is undetectable in α -cells from α -GsKO mice.

5) Why didn't the authors stimulate cells in Figure 5d and 5e with the adenosine receptor agonist?

We carried out these experiments. We incorporated the following text into the revised manuscript (page 19, top):

"To further corroborate the involvement of α -cell G_s signaling in A2AR-stimulated glucagon secretion, we treated control islets and islets derived from α -GsKO mice (α -GsKO islets) with the A2AR agonist UK 432097 (50 nM) (Fig. 3f, g). We found that UK 432097-induced glucagon secretion was absent or nearly completely abolished in α -GsKO islets at both G3 and G12, respectively. In addition, we carried out vivo experiments with PSB 0777 (1 mg/kg, i.p.), another selective A2AR agonist⁸. In agreement with the in vitro data (Fig. 3f, g), PSB 0777-induced increases in plasma glucagon levels were abolished in α -GsKO mice, as compared to control littermates (Supplementary Fig. 6c). These data clearly indicate that A2AR-stimulated glucagon secretion requires α -cell G_s signaling."

6) In figure 6, why aren't there homeostatic alterations in insulin levels when glucagon levels are reduced?

The reviewer raises a very interesting point. To address this issue, we incorporated the following text into the Discussion section of the revised manuscript (**page 27, top**): "Although α -GsKO mice showed a significant decrease in plasma glucagon levels (Fig. 6a, d), this deficit did not cause any significant changes in plasma insulin levels (Fig. 6b, e). We speculate that chronic hypoglucagonemia causes compensatory changes involving other factors and neuronal pathways that can maintain normal plasma insulin levels. In agreement with this notion, previous studies demonstrated that the near-total ablation of α -cells or suppression of α -cell glucagon expression does not have any discernible effect on plasma insulin levels in vivo¹²⁻¹⁴. In contrast, acute lowering of plasma glucagon levels due to activation of an inhibitory DREADD expressed in mouse α -cells led to impaired insulin release in vivo¹⁵."

Reviewer #3

The manuscript entitled "Novel mouse models establish a key metabolic role for α -cell Gs signaling" by Liu et. al demonstrate that Gs-coupled signaling impacts α -cell function. Understanding alpha cell function and the regulation of glucagon secretion could have important implications for diabetes treatment so the topic is timely and relevant. The authors use chemogenetics to activate Gqs signaling within Gcg cells in lean and obese animals and study in vivo and ex vivo glucagon and insulin secretion. DREADD activation in both lean and obese animals stimulated glucagon and insulin secretion and improved IP glucose tolerance. The authors also employ several mouse models that disrupt Gs signaling (α -GsD and Gas) or Gs adenosine signaling (A2AR) and show that the A2A adenosine receptors play a role in stimulating glucagon secretion under low glucose conditions. Although there are interpretive issues, the authors state that the perfusion studies agree. Long-term disruption of Gas signaling lowers Gcg expression, islet glucagon content, and consequently basal and glucagon responses to various stimuli. The authors conclude that Gas is a novel target for regulating glucagon secretion. Although there is enthusiasm for the topic of research, there are weaknesses associated with the mouse model, the statistical analysis, and some issues with data interpretation.

Thank you for all your helpful comments which have led to a greatly improved revised manuscript.

Major comments:

1. The Gcg Cre utilized here has not been validated for brain expression. Gcg neurons are known to be expressed in the hindbrain and function to regulate various physiological functions including feeding patterns, body weight in some cases, motivation for feeding etc. More and more studies are being published on the impact of these neurons on physiology. Therefore, validation that the genetic manipulation has not targeted the brain is required. They easily could have done the HA tag staining should have been completed in the nucleus of the solitary tract- why it was done in the hypothalamus instead is unclear.

The reviewer raises an important point. As requested, we studied whether α -GsD mice also express the HA-tagged GsD designer receptor in the region of the nucleus of the solitary tract (NTS region). The use of an anti-HA antibody that we routinely use to visualize the expression of HA-tagged DREADDs in the brain (see, for example, Fig 1b in ref. PMID: 26743492) failed to yield a detectable immunofluorescence signal in the NTS region, indicative of the lack of GsD expression in the NTS or of very low GsD expression levels below the detection threshold of the protocol used.

Please see the new Supplementary Fig. 1c in the revised version of the manuscript. We also added the following short paragraph to the Results section (page 7, top): "Moreover, immunofluorescence staining of brain slices prepared from α -GsD mice failed to detect GsD expression in proglucagon-producing neurons of the nucleus tractus solitarius (NTS; Supplementary Fig. 1c)."

2. Were the data in Figure 1 analyzed with an ANOVA for repeated measures? If there is a group effect, but no time effect then statistically you can't say that DCZ actually increased glucagon or insulin. The statistics need to be redone with an ANOVA for repeated measures with an appropriate post hoc analysis for all graphs where time is a variable.

Yes, all data shown in Fig. 1a-f were analyzed via two-way repeated measure ANOVA for time and group with Bonferroni correction for pairwise post hoc comparisons of time. In Fig. 1d-f, we examined group difference at each time point separately because there was significant interaction between group and time. Moreover, we subjected all other data presented in this manuscript to a thorough statistical analysis following the guidance of the NIDDK Biostatistics Program.

3. Figure 1: Although it is not clear whether this is statistically significant, there is a 50% drop in glucagon with saline injection and with DCZ in the control animals. This drop is prevented in the GsD animals with DCZ-again it is not clear whether DCZ actually significantly increases glucagon at 15 min or whether it is simply preventing a drop in glucagon. This finding is very curious and needs further exploration/explanation.

As requested by the reviewer, the data shown in Fig. 1a-f were subjected to a thorough statistical analysis, following the advice of the NIDDK Biostatistics Program. Details regarding the statistical tests used are given in the figure legends.

Based on the outcome of these analyses, we modified the text as follows ("Stimulation of α -cell G_s signaling leads to hyperglucagonemia and hyperinsulinemia in vivo") (page 7, center): "Similar to previous observations¹⁵, i.p. injection of control and α -GsD mice with saline or of control mice with DCZ resulted in reduced plasma glucagon levels 15 and 30 min after injection ($p < 0.0001$; one-way repeated measures ANOVA, followed by post-hoc Bonferroni adjustment) (Fig. 1a, d). Although the precise mechanism underlying this phenomenon remains unclear, we speculate that the increase in blood glucose levels (see below; Fig. 1c, f) caused by the injection stress impairs glucagon secretion from α -cells. Although there was a trend towards higher basal plasma glucagon level in α -GsD mice (time 0; Fig. 1d), this effect failed to reach statistical significance ($p = 0.085$; Student's t-test at time 0). Importantly, DCZ treatment of α -GsD mice led to a statistically significant increase in plasma glucagon levels (Fig. 1d), thus overcoming the inhibitory effect on glucagon release caused by the injection stress."

4. Figure 1: Is there a statistical increase in glucose with the saline or DCZ injection? If so, then these studies are likely confounded by the animal's stress levels.

As requested by the reviewer, the data shown in Fig. 1a-f were subjected to a thorough statistical analysis, following the advice of the NIDDK Biostatistics Program. Details regarding the statistical tests used are given in the figure legends.

Saline treatment of α -GsD mice and control littermates resulted in statistically significant increases in blood glucose levels ($p < 0.0001$ at 15 and 30 min after injection (time 0); one-way repeated measures with time ANOVA for each group) (Fig. 1c). While this effect persisted in DCZ-treated control mice, blood glucose levels remained unchanged after DCZ injection of α -

GsD mice (Fig. 1f), most likely due the increase in plasma insulin levels observed with DCZ-treated α -GsD mice (Fig. 1e) that "neutralized" the hyperglycemic effect of the injection stress. To address these points, we added additional text to the Results section (page 8, top).

5. Figure 1 and Supp Figure 1, perfusion studies: Is the relative increase in glucagon with the Ala or AAM greater than the control animals? It seems not. Thus, the increase in glucagon is simply due to an increase in basal concentrations. If this is the case then there seems to be an ex vivo and in vivo difference in glucagon responses that needs to be discussed. Further, it is unclear why a doubling of glucagon under "basal conditions" is visible in perfused islets but not in vivo. The fact that the insulin is normal at baseline and is higher with amino acid stimulus even though the relative increase in glucagon is similar dissociates glucagon and insulin responses to amino acids under these conditions. This becomes hard to see with the expanded scale in Figure 1g. The components of the perfusion experiments should be separated out so the data can be visualized on an appropriate scale for Fig 1g. In fact, is it the scale or did the control alpha cells not increase glucagon secretion in response to alanine?

As requested by the reviewer, we subjected the perfusion data shown in Fig. 1 and Suppl. Fig. 1 to a thorough statistical analysis, following the advice of the NIDDK Biostatistics Group. Details regarding the statistical tests used are given in the figure legends.

Question: "Is the relative increase in glucagon with the Ala or AAM greater than the control animals?"

To correct for differences in basal hormone secretion, we calculated area of the curve (AOC) values by subtracting the areas under or over the baseline³. This type of analysis is the method of choice when baseline levels between two or more experimental groups differ³. In the presence of alanine, DCZ treatment of α -GsD islets caused a statistically significant increase in glucagon release at both G12 and G3, as compared to control islets (Fig. 1g, h in the revised manuscript). We also subjected the insulin secretion data shown in Fig. 1h of the original manuscript to the same type of analysis. We found that insulin responses were significantly elevated at G12 (but not at G3) in α -GsD islets, as compared to control islets (new Fig. 1i, j). In the absence of DCZ, amino acid-induced glucagon and insulin release were similar in α -GsD and control islets (revised version of Suppl. Fig. 1k, l). We made appropriate changes in the revised manuscript.

Question: "Further, it is unclear why a doubling of glucagon under "basal conditions" is visible in perfused islets but not in vivo."

As stated by the reviewer, basal glucagon release at both G3 and G12 was significantly higher in α -GsD islets, as compared to control islets (Fig. 1g and Supplementary Fig. 1k). This observation is consistent with previous findings that the GsD designer receptor shows a certain degree of constitutive activity under distinct experimental conditions^{11,16}. As pointed out in the discussion section of the revised manuscript, previous studies have shown that the GsD designer receptor shows a certain degree of constitutive activity under distinct experimental conditions^{11,16}. Most likely, this increase in ligand-independent signaling is responsible for the observation that basal glucagon secretion is elevated in α -GsD islets. In contrast, in vivo studies showed that basal plasma glucagon levels were not significantly different between α -GsD mice and control littermates (Fig. 1a, d). It is likely that chronic activation of α -cell G_s caused by GsD-mediated constitutive signaling triggers compensatory changes in vivo that can maintain normal plasma glucagon levels and euglycemia. In agreement with this notion, circulating glucagon levels are regulated by various neuronal pathways and other factors (e.g. hormones and other circulating bioactive agents) that are not present in isolated islets^{17,18}.

We address this issue in the Discussion section of the revised manuscript (page 24, top). We also made changes in the Results section (page 8, bottom).

Question: "In fact, is it the scale or did the control alpha cells not increase glucagon secretion in response to alanine?"

Thank you raising this point. The scale used in the original version of Fig. 1g did not reveal the small increase in glucagon secretion triggered by treatment of control islets with alanine. To better visualize this response, we added an insert focusing on alanine-stimulated glucagon secretion in the revised version of Fig. 1g (for a statistical analysis of the data, please see Fig. 1h).

6. Statistical analysis of the perfusion studies needs to be reported.

As requested by the reviewer, we subjected all islet perfusion data to a thorough statistical analysis. Statistical tests were chosen based on the advice of the NIDDK Biostatistics Group. Details regarding the statistical tests used are given in the figure legends. To quantitate changes in hormone levels, we calculated AOC values that correct for altered hormone baseline levels³. This type of analysis is the method of choice when baseline levels between two or more experimental groups differ³.

7. Body weight of Adora2af/f Gcg cre mice is needed.

We added body weight data to Suppl. Fig. 4 (new panel 'g'). Body weights were similar in control and mutant mice.

8. Interpretation of the 2dg data in the alpha-A2AR KO mice is not accurate. These mice had decreased glucagon and increased glucose responses to 2DG. Under glucoprivic conditions, increases in glucagon function to stimulate hepatic glucose production and increase plasma glucose. Since insulin levels are the same, then this clearly indicates that the insulin to glucagon ratio should favor a suppression of hepatic glucose production. Therefore, some other counterregulatory factor is regulating the glucose response to 2DG in the face of inadequate glucagon levels in these animals. This suggests off-target impact on sympathetic neurons. Plasma catecholamines, which also potentially respond to 2DG and regulate glucose homeostasis should be assessed.

Like the reviewer, we were also puzzled by these findings. For this reason, we decided to repeat this experiment with a new set of mice that we specifically generated to revisit this issue. Using this new batch of mice, we were unable to replicate the data presented in original version of the manuscript (Fig. 4g, h; probably an artifact). The new data show that 2-DG-induced increases in plasma glucagon, plasma insulin, and blood glucose levels were not significantly affected by the lack of α -cell A2ARs. We therefore replaced Fig. 4g, h (original manuscript) with a suppl. figure (new Suppl. Fig. 4q-s), displaying these new data. We also changed the text in the Results section accordingly in the revised manuscript (page 16, center).

9. There are more 2DG interpretive issues in Figure 7. Specifically in the second part of this sentence: "The glucopenia-induced increases in blood glucose levels were significantly more pronounced in a-GsKO mice (Fig. 7d), probably due to decreased plasma insulin levels (Supplementary Fig. 7a, b)." The 2dg-induced (and it's not 2DG it's the increase in glucose that drives the increase in insulin) insulin secretion was not reduced, it simply returned to baseline quicker. Thus, the conclusion that α -cell Gs signaling contributes to the counter-regulatory

glucagon responses is not accurate. This is another specific example where a repeated measures anova for time is necessary so that the responses over time can be accurately assessed and interpreted.

Thank you for these helpful comments. A two-way ANOVA for repeated measures with Bonferroni post hoc test revealed a significant group effect, although the time-by-group interaction was not statistically significant. Based on this analysis and the reviewer's comments, we rephrased the manuscript text as follows (page 21, center):
"In α -GsKO mice, the 2-DG-dependent increase in glucose-driven insulin secretion returned to baseline more rapidly than in control littermates (Supplementary Fig. 8a, b). It is likely that this effect is responsible for the elevated blood glucose levels observed with 2-DG-treated α -GsKO mice, as compared to 2-DG-treated control littermates (Fig. 7d)."

10. The interpretation of the glucagon responses in the alpha-GsKO mice is also curious. *Gcg* expression is decreased, islet glucagon content is decreased. So if there is limited gene and protein, then it is not surprising that there is no glucagon response to glucagon secretagogues. It also brings to question why the KO of alpha cell Gs leads to a decrease in *Gcg* expression in the first place.

Regarding the first part of the reviewer's comments:

As stated by the reviewer, *Gcg* RNA levels and glucagon content are markedly decreased (by ~70-80%) in α -GsKO islets, as compared to control islets. It is highly likely that these deficits make a major contribution to the functional impairments displayed by the α -GsKO islets. However, UK 432097- and isoproterenol-stimulated glucagon secretion was absent or nearly abolished in α -GsKO islets (revised Fig. 3f, g and Fig. 5d, e, respectively). These data indicate that the lack of α -cell G_s signaling also contributes to the functional deficits caused by α -cell G_s deficiency.

We now briefly discuss this issue in the revised version of the manuscript (page 27, bottom).

Regarding the second part of the reviewer's comments:

To investigate the mechanism underlying the decrease in *Gcg* expression caused by α -cell G_s deficiency, we carried out additional studies with cultured mouse α -cells (α -TC6 cells) and isolated mouse islets. Treatment of α -TC6 cells with PKI 14-22 (10 μ M), a highly selective inhibitor of PKA, a protein kinase activated by G_s signaling, led to significantly reduced *Gcg* RNA levels (new Supplementary Fig. 7g).

We now address this point in the revised version of the manuscript (page 20, center; page 27, bottom, and page 28, top). We also incorporated the following paragraph into the Discussion section:

"The activity of the rodent *Gcg* promoter is under the control of several regulatory factors, including a cAMP response element (CRE) (reviewed in^{19,20}. The promoter of the rodent *Gcg* gene contains a CRE sequence that is activated by cAMP-dependent protein kinase A, resulting in enhanced *Gcg* transcription^{21,22}. This observation is consistent with our finding that impaired α -cell G_s /cAMP signaling leads to reduced *Gcg* RNA levels (Fig. 5k) and decreased pancreatic glucagon content in α -GsKO islets (Fig. 5g, h). The latter finding also provides an explanation for the observation that treatment of α -GsKO islets with KCl or with an agonist that acts on G_q -coupled V1bRs resulted in impaired increases in glucagon secretion (Fig. 5f, 7h). In agreement with these published data^{21,22}, we showed that treatment of cultured mouse α -cells (TC6 cells) with PKI 14-22, a highly selective inhibitor of PKA, resulted in significantly decreased *Gcg* RNA levels (Supplementary Fig. 7g). In sum, these findings highlight the importance of basal α -cell G_s signaling in maintaining proper *Gcg* transcription and glucagon content in pancreatic islets."

11. It would be interested to know how somatostatin regulation of the alpha cell fits into this story.

The reviewer raises an interesting point. To explore this issue, we conducted several additional experiments. First, we demonstrated that DCZ treatment of α -GsD mice had no significant effect on plasma somatostatin (SS) levels, as compared to DCZ-treated control littermates (new Supplementary Fig. 1h). Moreover, pancreatic SST content was not affected by the lack of α -cell G_s signaling (new Supplementary Fig. 7d).

Previous studies have shown that SST release from δ -cells inhibits glucagon secretion from α -cells (reviewed in ref ²³). To investigate whether this effect was altered in α -GsKO islets, we incubated control and α -GsKO islets with a combination of two SST receptor inhibitors (PRL2915 + PRL3195, 1 μ M each) which are known to block the major SST receptor subtypes expressed by mouse α -cells (SSTR2, SSTR3, and SSTR5)²⁴. We monitored basal glucagon release as well as isoproterenol-induced glucagon secretion. The dramatic reductions in basal and isoproterenol-induced glucagon secretion caused by α -cell G_s deficiency in the absence of SST receptor blockade persisted in the presence of the SST receptor antagonists (new Supplementary Fig. 5f, g). These data suggest that it is unlikely that altered SST release from δ -cells contributes to the functional deficits displayed by α -GsKO mice or α -GsKO islets.

These new data have now been incorporated into the revised version of the manuscript (page 8, center; page 18, center; page 20, top).

Minor Comments:

12. Supp Fig 1, Panel B should show individual channels of all fluorophores used.

As requested by the reviewer, we added individual channels of all fluorophores used to Suppl. Fig. 1b.

References

1. Baron, M., *et al.* A Single-Cell Transcriptomic Map of the Human and Mouse Pancreas Reveals Inter- and Intra-cell Population Structure. *Cell Syst* **3**, 346-360 e344 (2016).
2. DiGrucio, M.R., *et al.* Comprehensive alpha, beta and delta cell transcriptomes reveal that ghrelin selectively activates delta cells and promotes somatostatin release from pancreatic islets. *Mol Metab* **5**, 449-458 (2016).
3. Virtue, S. & Vidal-Puig, A. GTTs and ITTs in mice: simple tests, complex answers. *Nat Metab* **3**, 883-886 (2021).
4. Hamilton, A., Eliasson, L. & Knudsen, J.G. Amino acids and the changing face of the α -cell. *Peptides* **166**, 171039 (2023).
5. El, K., *et al.* GIP mediates the incretin effect and glucose tolerance by dual actions on α cells and β cells. *Sci Adv* **7**(2021).
6. Yang, G.K., *et al.* Glucose decreases extracellular adenosine levels in isolated mouse and rat pancreatic islets. *Islets* **4**, 64-70 (2012).
7. Yu, Q., Shuai, H., Ahooghalandari, P., Gylfe, E. & Tengholm, A. Glucose controls glucagon secretion by directly modulating cAMP in alpha cells. *Diabetologia* **62**, 1212-1224 (2019).

8. El-Tayeb, A., *et al.* Development of Polar adenosine A_{2A} receptor agonists for inflammatory bowel disease: synergism with A_{2B} antagonists. *ACS Med Chem Lett* **2**, 890-895 (2011).
9. Szkudelski, T. & Szkudelska, K. Regulatory role of adenosine in insulin secretion from pancreatic β -cells--action via adenosine A₁ receptor and beyond. *J Physiol Biochem* **71**, 133-140 (2015).
10. Akhmedov, D., *et al.* Gs-DREADD knock-in mice for tissue-specific, temporal stimulation of cAMP signaling. *Mol Cell Biol* (2017).
11. Guettier, J.M., *et al.* A chemical-genetic approach to study G protein regulation of beta cell function in vivo. *Proc Natl Acad Sci U S A* **106**, 19197-19202 (2009).
12. Traub, S., *et al.* Pancreatic α Cell-Derived Glucagon-Related Peptides Are Required for β Cell Adaptation and Glucose Homeostasis. *Cell Rep* **18**, 3192-3203 (2017).
13. Thorel, F., *et al.* Normal glucagon signaling and beta-cell function after near-total alpha-cell ablation in adult mice. *Diabetes* **60**, 2872-2882 (2011).
14. Katoh, M.C., *et al.* MafB Is Critical for glucagon production and secretion in mouse pancreatic α Cells In Vivo. *Molecular Cell Biol* **38**(2018).
15. Zhu, L., *et al.* Intra-islet glucagon signaling is critical for maintaining glucose homeostasis. *JCI insight* **5**(2019).
16. Akhmedov, D., *et al.* Gs-DREADD knock-in mice for tissue-specific, temporal stimulation of cyclic AMP signaling. *Molecular Cell Biol* **37**(2017).
17. Andersen, D.B. & Holst, J.J. Peptides in the regulation of glucagon secretion. *Peptides* **148**, 170683 (2022).
18. Zhang, J., Zheng, Y., Martens, L. & Pfeiffer, A.F.H. The regulation and secretion of glucagon in response to nutrient composition: unraveling their intricate mechanisms. *Nutrients* **15**(2023).
19. Jin, T. Mechanisms underlying proglucagon gene expression. *J Endocrinology* **198**, 17-28 (2008).
20. Müller, T.D., *et al.* Glucagon-like peptide 1 (GLP-1). *Mol Metab* **30**, 72-130 (2019).
21. Knepel, W., Chafitz, J. & Habener, J.F. Transcriptional activation of the rat glucagon gene by the cyclic AMP-responsive element in pancreatic islet cells. *Molecular Cell Biol* **10**, 6799-6804 (1990).
22. Drucker, D.J., Campos, R., Reynolds, R., Stobie, K. & Brubaker, P.L. The rat glucagon gene is regulated by a protein kinase A-dependent pathway in pancreatic islet cells. *Endocrinology* **128**, 394-400 (1991).
23. Rorsman, P. & Huisling, M.O. The somatostatin-secreting pancreatic δ -cell in health and disease. *Nature reviews. Endocrinology* **14**, 404-414 (2018).
24. Svendsen, B. & Holst, J.J. Paracrine regulation of somatostatin secretion by insulin and glucagon in mouse pancreatic islets. *Diabetologia* **64**, 142-151 (2021).

Reviewers' comments:

Reviewer #1 (Remarks to the Author):

The revised manuscript is much improved and the authors have answered satisfyingly to most of the referee comments. However, this reviewer is concerned that the authors were unable to replicate some of the findings reported in the original version of the manuscript, when experiments were repeated as a consequence of comments from the referees (point 8 by both referee 1 and 3). No explanation for the discrepancy is provided other than “probably an artifact”. This is unsettling and the question arises whether also other experiments may be affected by similar artifacts.

Point 2. The immunofluorescence images are still not of good quality. The spatial resolution is worse than in the previous ms version and the glucagon, HA and DAPI stains are still very dim.

Minor:

The authors describe alpha-TC6 cells as “cultured mouse alpha cells” but that is misleading. The latter term should be reserved for primary mouse alpha cells. The tumor-derived clonal cell line alpha-TC6 should be named for what it is.

Lines 156-157. DZC should read DCZ.

Reviewer #2 (Remarks to the Author):

Thanks for your comprehensive and thoughtful responses to questions raised by the reviews.

Reviewer #3 (Remarks to the Author):

Comments to the authors:

While the authors have done much to address my concerns, there remain several issues of high concern with this manuscript. Some are noted here:

1. Regarding interpretations and my comments on Figure 1a, d:

The authors state:

Similar to previous observations¹⁵, i.p. injection of control and α -GsD mice with saline or of control mice with DCZ resulted in reduced plasma glucagon levels 15 and 30 min after injection ($p < 0.0001$; one-way repeated measures ANOVA, followed by post-hoc Bonferroni adjustment) (Fig. 1a, d). Although the precise mechanism underlying this phenomenon remains unclear, we speculate that the increase in blood glucose levels (see below; Fig. 1c, f) caused by the injection stress impairs glucagon secretion from α -cells.

This is not how stress physiology works. Glucagon is downstream of sympathetic activation (given the isoproterenol experiment the authors know this) and serves as a player in increasing hepatic glucose production. It is recommended to just simply state that “Although the precise mechanism underlying this phenomenon remains unclear, it is possible that the stress of the IP injection is confounding basal hormonal levels”.

However, the fact that the animals are stressed (and the authors admit to this) confounds the results of these studies. The authors should examine strategies to minimize stress as outlined by some of the many methods papers from the Vanderbilt MMPC.

2. Regarding comments on Figure 7d:

There are still no statistical comparisons for given time points reflected on the graph. If the 2-way ANOVA reveals a significant time x genotype interaction then the post hoc analysis will reveal at what time points this is significant. The time points where glucose higher in the alpha-GsKO mice needs to be denoted on the graph panel with an asterix over the time point.

3. In the results and in reference to the insulin levels after 2DG, please correct the line prior to the highlighted text which still incorrectly says "2-DG-induced insulin secretion....." . "At the same time, 2-DG-induced insulin secretion was also significantly reduced (Supplementary Fig. 8a, b). In α -GsKO mice, the 2-DG-dependent increase in glucose-driven insulin secretion returned to baseline more rapidly than in control littermates (Supplementary Fig. 8a, b). It is likely that this effect is responsible for the elevated blood glucose levels observed with 2-DG-treated α -GsKO mice, as compared to 2-DG-treated control littermates (Fig. 7d)."

4. There remains a huge concern over the conclusion that isoproterenol-stimulated glucagon secretion requires alpha-cell Gs signaling. The alpha-GsKO animals have limited Gcg gene or protein. How would UK 432097 stimulate glucagon if there is none there to stimulate? Putting it another way, what would the conclusion be if UK 432097 stimulated glucagon secretion? Where would the authors think that glucagon be coming from in that case?

03-27-2024

Response to the Referees' comments

NCOMMS-23-26486A, re-revised version

TITLE: " Novel mouse models establish a key metabolic role for α -cell G_s signaling"

by Liu Liu *et al.*

General response by the authors

We thank the reviewers for the careful review of our revised manuscript.

Two of my postdocs spent about 5 months carrying out the many additional experiments requested by the reviewers during the initial round of reviews. For these studies, they generated a large cohort of additional mutant mice. Clearly, our lab expanded an enormous amount of effort and resources to address the original comments of the three reviewers.

While reviewer 2 was fully satisfied with the revised version, reviewers 1 and 3 raised additional points. The points raised by reviewers 1 and 3 are valid; however, as outlined below, we were able to properly address all concerns of the two reviewers in the re-revised version of this manuscript.

We paid very close attention to the extra points raised by reviewers 1 and 3 and have made every attempt to address these comments in order to enhance the quality of the manuscript. Moreover, please note that most of these points are not central to the key conclusions that emerged from our work or can be explained in a rather straightforward fashion by the presented experimental *in vitro* and *in vivo* data.

Also, PLEASE NOTE: The key findings of this study were independently confirmed by the labs of Jurgen Wess (NIH) and Jonathan Campbell (Duke University). In fact, during a meeting that took place about two years ago, Jonathan and I realized that we had generated similar mouse models that yielded similar data. At this point, we decided to co-publish our data in *Nature Communications*.

We would like to reiterate that our joint efforts have led to many IMPORTANTTT NOVEL FINDINGS regarding the regulation of glucagon from pancreatic α -cells:

1. Selective stimulation of α -cell G_s signaling strongly promotes glucagon release *in vitro* and *in vivo*.
2. Selective stimulation of α -cell G_s signaling has beneficial metabolic effects on whole body glucose homeostasis.
3. Acute, selective activation of α -cell G_s signaling results in an insulinotropic effect that improves glucose homeostasis in both lean and obese, glucose-intolerant mice.
4. Our findings strongly support the novel concept that α -cell G_s signaling plays a key role in stimulating sufficient glucagon release under hypoglycemic conditions.
5. Our data strongly suggest that islet adenosine acts as an autocrine or paracrine factor that promotes glucagon release by activating the α -cell A_{2A} receptor/ G_s signaling cascade when glucose levels are low.

6. Disruption of α -cell G_s signaling suppresses the expression of the *Gcg* gene, leading to reduced glucagon synthesis and storage. This deficit is most likely responsible for the hypoglucagonemia phenotype displayed by the α -GsKO mice. These findings highlight the importance of basal α -cell G_s signaling in maintaining proper *Gcg* transcription and glucagon content in pancreatic islets.
7. We also present data strongly suggesting that α -cell G_s signaling is required for the proper release of glucagon and insulin after consumption of a mixed meal.
8. We demonstrate that α -cell G_s signaling plays an important role in mediating the counter-regulatory stimulation of glucagon secretion during hypoglycemic and glucopenic states.

Comments by Reviewer #1

Point 1.

The revised manuscript is much improved and the authors have answered satisfyingly to most of the referee comments. However, this reviewer is concerned that the authors were unable to replicate some of the findings reported in the original version of the manuscript, when experiments were repeated as a consequence of comments from the referees (point 8 by both referee 1 and 3). No explanation for the discrepancy is provided other than “probably an artifact”. This is unsettling and the question arises whether also other experiments may be affected by similar artifacts.

OUR RESPONSE

We are glad that the reviewer considers our manuscript much improved (after a very positive initial review), stating that “the authors have answered satisfyingly to most of the referee comments.”

The specific experiment that the reviewer is referring to deals with the following observation, as described by rev. 1 in her/his original review (point 8):

“P 15/ Fig 4g,h: 2-DG administration triggers glucagon secretion, which is impaired in the α -A2AR-KO. Yet, the resulting hyperglycemia is more pronounced in the KO, despite similar insulin levels. This seemingly confusing observation deserves to be discussed.”

Like the reviewer, we were also puzzled by these data since they were not consistent with all other experimental data contained in the manuscript. For this reason, we decided to repeat this particular experiment. The repeat experiment showed that the effect of 2-DG treatment on hyperglycemia was similar in α -A2AR-KO and control mice (Suppl. Fig. 4q-s in the revised manuscript). Please note that this experimental outcome is not central to the key conclusions outlined above.

We would also like to note that the outcome of *in vivo* studies in mice is affected by numerous external parameters (e.g. mouse housing conditions, litter size, precise time of the day when a specific experiment is done, temperature and humidity in the animal facility, prior presence of technical staff in the animal room, etc.) which occasionally leads to unexpected results. My lab has more than 25 years of experience in analyzing metabolic parameters in mutant mice *in vivo*. Our manuscript describes a very large number of *in vivo* experiments. For this reason, the likelihood that a specific experiment (which, in this case, is not critical for the

key conclusions that can be drawn from this study) is not fully reproducible increases (p values only indicate a certain likelihood that an observed difference is 'real'). In this particular case that involved very "puzzling data", we wondered whether the observed difference might have been a "chance occurrence", as we confirmed in the repeat experiment.

Please note that the key conclusions that can be drawn from our study are mutually supported by the parallel use of in vivo and in vitro approaches. I would also like to emphasize that other labs have always been able to reproduce our published data in the past. I take particular pride in the fact that my lab is known for its exceptional scientific rigor.

Point 2.

The immunofluorescence images are still not of good quality. The spatial resolution is worse than in the previous ms version and the glucagon, HA and DAPI stains are still very dim.

OUR RESPONSE

The reviewer refers to the immunofluorescence images in Suppl. Fig. 1b in the revised version of the manuscript (point 2 in the original review). The quality of these images may have suffered from the PDF conversion during the submission process. To address this point, we carried out additional immunofluorescence experiments that yielded images of vastly improved quality (new Suppl Fig. 1b). We modified the legend of the new version of Suppl. Fig. 1b as follows:

"(b) Immunofluorescence staining of pancreatic slices from α -GsD mice and control mice. The HA-tagged GsD receptor was stained with an anti-HA antibody (Alexa Fluor, green). Pancreatic α -cells were visualized with an anti-glucagon antibody (Alexa Fluor, red), and β -cells were stained with an anti-insulin antibody (Alex Fluor, red), respectively. Nuclei were stained blue with DAPI mounting medium. Contrast was adjusted for improved visualization. Cells expressing HA-tagged GsD receptors (green stain) were not detectable in control pancreatic islets (upper two rows). In contrast, studies with islets from α -GsD mice showed that glucagon-containing α -cells, but not insulin-producing β -cells, expressed GsD designer receptors (green stain) on their cell surface (lower two rows). Column 4 shows enlarged images of the areas highlighted in column 3. Scale bars: columns 1-3: 50 μ m; column 4: 10 μ m."

Minor:

The authors describe alpha-TC6 cells as "cultured mouse alpha cells" but that is misleading. The latter term should be reserved for primary mouse alpha cells. The tumor-derived clonal cell line alpha-TC6 should be named for what it is.

OUR RESPONSE

As requested by the reviewer, we made this change in the re-revised version of the manuscript. See page 20, center.

Lines 156-157. DZC should read DCZ.

OUR RESPONSE

Thank you for pointing out this error. We made this change in the re-revised version of the manuscript. See page 8, 1st paragraph.

Comments by Reviewer #3

While the authors have done much to address my concerns, there remain several issues of high concern with this manuscript. Some are noted here:

OUR RESPONSE

We are glad that the reviewer acknowledges that we have "done much" to address her/his concerns.

1. Regarding interpretations and my comments on Figure 1a, d:

The authors state:

Similar to previous observations¹⁵, i.p. injection of control and α -GsD mice with saline or of control mice with DCZ resulted in reduced plasma glucagon levels 15 and 30 min after injection ($p < 0.0001$; one-way repeated measures ANOVA, followed by post-hoc Bonferroni adjustment) (Fig. 1a, d). Although the precise mechanism underlying this phenomenon remains unclear, we speculate that the increase in blood glucose levels (see below; Fig. 1c, f) caused by the injection stress impairs glucagon secretion from α -cells.

This is not how stress physiology works. Glucagon is downstream of sympathetic activation (given the isoproterenol experiment the authors know this) and serves as a player in increasing hepatic glucose production. It is recommended to just simply state that "Although the precise mechanism underlying this phenomenon remains unclear, it is possible that the stress of the IP injection is confounding basal hormonal levels".

OUR RESPONSE

We fully agree with the reviewer. We rephrased this sentence in the re-revised version of the manuscript (page 7, bottom paragraph).

However, the fact that the animals are stressed (and the authors admit to this) confounds the results of these studies. The authors should examine strategies to minimize stress as outlined by some of the many methods papers from the Vanderbilt MMPC.

OUR RESPONSE

Treatment of mice with an i.p. injection represents a stressor that cannot be avoided (but mitigated; see below). All animals were kept under conditions that minimized stress (proper housing conditions, enrichment strategies, etc.), according to the Guidelines of the NIH Animal Research Advisory Committee (ARAC). These guidelines closely reflect the recommendations of the Vanderbilt MMPC. I would also like to mention that I serve on the Animal Care and Use Committee of NIDDK. For this reason, I am very familiar with the ARAC guidelines regarding minimizing stress in experimental animals.

Moreover, to reduce injection-induced stress, we handled mice daily for one week including i.p. vehicle injections prior to performing actual experiments involving i.p. injections. We now mention this fact in the re-revised version of the manuscript (page 32, bottom). Despite all these stress-relieving measures, the injection-induced decrease in plasma glucagon levels was still detectable, in agreement with published data (PMID: 31012868; PMID: 34752420).

We also sought the advice of several other PIs who routinely measure plasma glucagon levels in mice following i.p. injections with vehicle. Most individuals we contacted described similar injection effects on plasma glucagon levels in mice.

To clarify this point, we added additional text to the Methods section (page 32, bottom).

As suggested by Reviewer 1, we reworded the text dealing with this matter as follows:
"Although the precise mechanism underlying this phenomenon remains unclear, it is possible that the stress of the i.p. injection is confounding basal hormonal levels." Page 7, bottom paragraph.

2. Regarding comments on Figure 7d:

There are still no statistical comparisons for given time points reflected on the graph. If the 2-way ANOVA reveals a significant time x genotype interaction then the post hoc analysis will reveal at what time points this is significant. The time points where glucose higher in the alpha-GsKO mice needs to be denoted on the graph panel with an asterix over the time point.

OUR RESPONSE

As requested by the reviewer, we now added a p value (0.0289) to the 120 min time point (no other time points showed statistically significant differences; Fig. 7d). The *Nature Commun* instructions for authors recommend showing actual p values in the figures, rather than asterisks.

3. In the results and in reference to the insulin levels after 2DG, please correct the line prior to the highlighted text which still incorrectly says "2-DG-induced insulin secretion....." .
"At the same time, 2-DG-induced insulin secretion was also significantly reduced (Supplementary Fig. 8a, b). In α -GsKO mice, the 2-DG-dependent increase in glucose-driven insulin secretion returned to baseline more rapidly than in control littermates (Supplementary Fig. 8a, b). It is likely that this effect is responsible for the elevated blood glucose levels observed with 2-DG-treated α -GsKO mice, as compared to 2-DG-treated control littermates (Fig. 7d)."

OUR RESPONSE

We apologize for this oversight. As requested by the reviewer, we rephrased this sentence in the re-revised version of the manuscript (page 21, center paragraph).

4. There remains a huge concern over the conclusion that isoproterenol-stimulated glucagon secretion requires alpha-cell Gs signaling. The alpha-GsKO animals have limited Gcg gene or protein. How would UK 432097 stimulate glucagon if there is none there to stimulate? Putting it another way, what would the conclusion be if UK 432097 stimulated glucagon secretion? Where would the authors think that glucagon be coming from in that case?

OUR RESPONSE

The reviewer raises an important point that we should have addressed in more detail in the revised version of the manuscript. The following observations/comments should be able to allay the reviewer's concerns:

The β -adrenergic, A_{2A} adenosine, and GIP receptors are G_s -coupled receptors (PMID: 38123151). We mention this fact for the β -adrenergic and A_{2A} adenosine receptors in the main text of the manuscript (page 5, first paragraph; page 11, 1st paragraph; page 17, line 9; page 24, 2nd paragraph). Moreover, for increased clarity, we added the following sentence on page 21 (4th and 5th line from the bottom): "Previous work has shown that the GIP receptor is selectively linked to G_s (ref. 60)".

Thus, as expected, studies with isolated islets showed that the lack of α -cell G_{α_s} virtually abolished the ability of GPCR agonists acting on α -cell β -adrenergic, A_{2A} adenosine, and GIP

receptors (isoproterenol, UK 432097, and GIP, respectively) to promote glucagon release (Figure 5d, e; 4b, 7h). In contrast, treatment of α -GsKO islets with a V1bR agonist (d[Leu⁴,Lys⁸]VP) still resulted in a significant stimulation of glucagon secretion (Figure 5f; G12). In contrast to the β -adrenergic, A_{2A}, and GIP receptors, the V1bR is a G_{q/11}-coupled receptor that releases glucagon by activating α -cell G_{q/11} signaling (PMID: 34752420). The ability of the V1bR to promote significant glucagon secretion in the absence of α -cell G α_s clearly indicates that there is still a substantial amount of releasable glucagon left in α -GsKO mice (Figure 5h). Moreover, KCl retained the ability to stimulate glucagon secretion in α -GsKO islets, although with reduced efficacy (see, for example, Figure 5f, 7h). In agreement with the in vitro data, α -GsKO mice retained the ability to stimulate the release of glucagon in vivo under glucopenic conditions, although this response was reduced relative to control littermates (Fig. 7c). Finally, pancreata from α -Gs KO mice still contain a considerable amount of glucagon (Figure 5h) which can serve as the source of glucagon in response to non-G_s-dependent stimuli.

These observations clearly indicate that the relative inability of GPCR agonists acting on α -cell β -adrenergic, A_{2A}, and GIP receptors to promote glucagon release from α -Gs KO islets is due to the selective inactivation of α -cell G_s signaling, in combination with the reduced amount of glucagon stored by α -Gs KO islets.

We added an extra paragraph to the Discussion section of the re-revised version of the manuscript to clarify this matter (page 26, bottom and page 27, top).

REVIEWERS' COMMENTS

Reviewer #1 (Remarks to the Author):

This reviewer remains overall enthusiastic about the manuscript but react to the apparent lack of concern that data were not reproducible. I agree that this particular experiment is not critical for the key conclusions. Let us hope that the authors' scientific rigor has prevented any further instances of "chance occurrences" that make data difficult to reproduce. Suppl Fig 1b is ok. The authors may want to adjust the position of the dashed square in the merged image to match the enlarged area. There seems to be a shift in the bottom row.

Reviewer #4 (Remarks to the Author):

The authors have thoughtfully responded to all reviewer comments, I have no further suggestions.

05-29-2024

Response to the comments of the reviewers

NCOMMS-23-26486B-Z, final version

NEW TITLE: "Intra-islet α -cell Gs signaling promotes glucagon release"
by Liu Liu *et al.*

REVIEWERS' COMMENTS

Reviewer #1 (Remarks to the Author)

This reviewer remains overall enthusiastic about the manuscript but react to the apparent lack of concern that data were not reproducible. I agree that this particular experiment is not critical for the key conclusions. Let us hope that the authors' scientific rigor has prevented any further instances of "chance occurrences" that make data difficult to reproduce.

Suppl Fig 1b is ok. The authors may want to adjust the position of the dashed square in the merged image to match the enlarged area. There seems to be a shift in the bottom row.

OUR RESPONSE: Thank you for positive response. We modified Suppl. Fig. 1b as requested by the reviewer.

Reviewer #4 (Remarks to the Author)

The authors have thoughtfully responded to all reviewer comments, I have no further suggestions.

OUR RESPONSE: Thank you for positive response.